# A unified variance-reduced accelerated gradient method for convex optimization

**Guanghui Lan**
H. Milton Stewart School of Industrial & Systems Engineering
Georgia Institute of Technology
Atlanta, GA 30332
george.lan@isye.gatech.edu

**Zhize Li**
Institute for Interdisciplinary Information Sciences
Tsinghua University
Beijing 100084, China
zz-li14@mails.tsinghua.edu.cn

**Yi Zhou**
IBM Almaden Research Center
San Jose, CA 95120
yi.zhou@ibm.com

## Abstract

We propose a novel randomized incremental gradient algorithm, namely, VAriance-Reduced Accelerated Gradient (Varag ), for finite-sum optimization. Equipped with a unified step-size policy that adjusts itself to the value of the condition number, Varag exhibits the unified optimal rates of convergence for solving smooth convex finite-sum problems directly regardless of their strong convexity. Moreover, Varag is the first accelerated randomized incremental gradient method that benefits from the strong convexity of the data-fidelity term to achieve the optimal linear convergence. It also establishes an optimal linear rate of convergence for solving a wide class of problems only satisfying a certain error bound condition rather than strong convexity. Varag can also be extended to solve stochastic finite-sum problems.

## 1 Introduction

The problem of interest in this paper is the convex programming (CP) problem given in the form of

$$\psi^* := \min_{x \in X} \left\{ \psi(x) := \tfrac{1}{m}\sum_{i=1}^m f_i(x) + h(x) \right\}. \tag{1.1}$$

Here, $X \subseteq \mathbb{R}^n$ is a closed convex set, the component function $f_i : X \to \mathbb{R}, \ i = 1, \ldots, m$, are smooth and convex function with $L_i$-Lipschitz continuous gradients over $X$, i.e., $\exists L_i \geq 0$ such that

$$\|\nabla f_i(x_1) - \nabla f_i(x_2)\|_* \leq L_i \|x_1 - x_2\|, \ \forall x_1, x_2 \in X, \tag{1.2}$$

and $h : X \to \mathbb{R}$ is a relatively simple but possibly nonsmooth convex function. For notational convenience, we denote $f(x) := \tfrac{1}{m}\sum_{i=1}^m f_i(x)$ and $L := \tfrac{1}{m}\sum_{i=1}^m L_i$. It is easy to see that $f$ has $L$-Lipschitz continuous gradients, i.e., for some $L_f \geq 0$, $\|\nabla f(x_1) - \nabla f(x_2)\|_* \leq L_f \|x_1 - x_2\| \leq L\|x_1 - x_2\|, \ \forall x_1, x_2 \in X$. It should be pointed out that it is not necessary to assume $h$ being strongly convex. Instead, we assume that $f$ is possibly strongly convex with modulus $\mu \geq 0$.

We also consider a class of stochastic finite-sum optimization problems given by

$$\psi^* := \min_{x \in X} \left\{ \psi(x) := \tfrac{1}{m}\sum_{i=1}^m \mathbb{E}_{\xi_i}[F_i(x, \xi_i)] + h(x) \right\}, \tag{1.3}$$

where $\xi_i$'s are random variables with support $\Xi_i \subseteq \mathbb{R}^d$. It can be easily seen that (1.3) is a special case of (1.1) with $f_i = \mathbb{E}_{\xi_i}[F_i(x, \xi_i)], i = 1, \ldots, m$. However, different from deterministic finite-sum optimization problems, only noisy gradient information of each component function $f_i$ can be accessed for the stochastic finite-sum optimization problem in (1.3). Particularly, (1.3) models the generalization risk minimization in distributed machine learning problems.

Finite-sum optimization given in the form of (1.1) or (1.3) has recently found a wide range of applications in machine learning (ML), statistical inference, and image processing, and hence becomes the subject of intensive studies during the past few years. In centralized ML, $f_i$ usually denotes the loss generated by a single data point, while in distributed ML, it may correspond to the loss function for an agent $i$, which is connected to other agents in a distributed network.

Recently, randomized incremental gradient (RIG) methods have emerged as an important class of first-order methods for finite-sum optimization (e.g.,[5, 14, 27, 9, 24, 18, 1, 2, 13, 20, 19]). In an important work, [24] (see [5] for a precursor) showed that by incorporating new gradient estimators into stochastic gradient descent (SGD) one can possibly achieve a linear rate of convergence for smooth and strongly convex finite-sum optimization. Inspired by this work, [14] proposed a stochastic variance reduced gradient (SVRG) which incorporates a novel stochastic estimator of $\nabla f(x_{t-1})$. More specifically, each epoch of SVRG starts with the computation of the exact gradient $\tilde{g} = \nabla f(\tilde{x})$ for a given $\tilde{x} \in \mathbb{R}^n$ and then runs SGD for a fixed number of steps using the gradient estimator

$$G_t = (\nabla f_{i_t}(x_{t-1}) - \nabla f_{i_t}(\tilde{x})) + \tilde{g},$$

where $i_t$ is a random variable with support on $\{1, \ldots, m\}$. They show that the variance of $G_t$ vanishes as the algorithm proceeds, and hence SVRG exhibits an improved linear rate of convergence, i.e., $\mathcal{O}\{(m + L/\mu) \log(1/\epsilon)\}$, for smooth and strongly convex finite-sum problems. See [27, 9] for the same complexity result. Moreover, [2] show that by doubling the epoch length SVRG obtains an $\mathcal{O}\{m \log(1/\epsilon) + L/\epsilon\}$ complexity bound for smooth convex finite-sum optimization.

Observe that the aforementioned variance reduction methods are not accelerated and hence they are not optimal even when the number of components $m = 1$. Therefore, much recent research effort has been devoted to the design of optimal RIG methods. In fact, [18] established a lower complexity bound for RIG methods by showing that whenever the dimension is large enough, the number of gradient evaluations required by any RIG methods to find an $\epsilon$-solution of a smooth and strongly convex finite-sum problem i.e., a point $\bar{x} \in X$ s.t. $\mathbb{E}[\|\bar{x} - x^*\|_2^2] \leq \epsilon$, cannot be smaller than

$$\Omega\left(\left(m + \sqrt{\tfrac{mL}{\mu}}\right) \log \tfrac{1}{\epsilon}\right). \tag{1.4}$$

As can be seen from Table 1, existing accelerated RIG methods are optimal for solving smooth and strongly convex finite-sum problems, since their complexity matches the lower bound in (1.4).

Notwithstanding these recent progresses, there still remain a few significant issues on the development of accelerated RIG methods. Firstly, as pointed out by [25], existing RIG methods can only establish accelerated linear convergence based on the assumption that the regularizer $h$ is strongly convex, and fails to benefit from the strong convexity from the data-fidelity term [26]. This restrictive assumption does not apply to many important applications (e.g., Lasso models) where the loss function, rather than the regularization term, may be strongly convex. Specifically, when dealing with the case that only $f$ is strongly convex but not $h$, one may not be able to shift the strong convexity of $f$, by subtracting and adding a strongly convex term, to construct a simple strongly convex term $h$ in the objective function. In fact, even if $f$ is strongly convex, some of the component functions $f_i$ may only be convex, and hence these $f_i$s may become nonconvex after subtracting a strongly convex term. Secondly, if the strongly convex modulus $\mu$ becomes very small, the complexity bounds of all existing RIG methods will go to $+\infty$ (see column 2 of Table 1), indicating that they are not robust against problem ill-conditioning. Thirdly, for solving smooth problems without strong convexity, one has to add a strongly convex perturbation into the objective function in order to gain up to a factor of $\sqrt{m}$ over Nesterov's accelerated gradient method for gradient computation (see column 3 of Table 1). One significant difficulty for this indirect approach is that we do not know how to choose the perturbation parameter properly, especially for problems with unbounded feasible region (see [2] for a discussion about a similar issue related to SVRG applied to non-strongly convex problems). However, if one chose not to add the strongly convex perturbation term, the best-known complexity would be given by Katyusha[ns][1], which are not more advantageous over Nesterov's orginal method. In other words, it does not gain much from randomization in terms of computational complexity.

Finally, it should be pointed out that only a few existing RIG methods, e.g., RGEM[19] and [16], can be applied to solve stochastic finite-sum optimization problems, where one can only access the stochastic gradient of $f_i$ via a stochastic first-order oracle (SFO).

Table 1: Summary of the recent results on accelerated RIG methods

| Algorithms | Deterministic smooth strongly convex | Deterministic smooth convex |
|---|---|---|
| RPDG[18] | $\mathcal{O}\left\{(m+\sqrt{\frac{mL}{\mu}})\log\frac{1}{\epsilon}\right\}$ | $\mathcal{O}\left\{(m+\sqrt{\frac{mL}{\epsilon}})\log\frac{1}{\epsilon}\right\}$[1] |
| Catalyst[20] | $\mathcal{O}\left\{(m+\sqrt{\frac{mL}{\mu}})\log\frac{1}{\epsilon}\right\}$[1] | $\mathcal{O}\left\{(m+\sqrt{\frac{mL}{\epsilon}})\log^2\frac{1}{\epsilon}\right\}$[1] |
| Katyusha[1] | $\mathcal{O}\left\{(m+\sqrt{\frac{mL}{\mu}})\log\frac{1}{\epsilon}\right\}$ | $\mathcal{O}\left\{(m\log\frac{1}{\epsilon}+\sqrt{\frac{mL}{\epsilon}})\right\}$[1] |
| Katyusha[ns][1] | NA | $\mathcal{O}\left\{\frac{m}{\sqrt{\epsilon}}+\sqrt{\frac{mL}{\epsilon}}\right\}$ |
| RGEM[19] | $\mathcal{O}\left\{(m+\sqrt{\frac{mL}{\mu}})\log\frac{1}{\epsilon}\right\}$ | NA |

**Our contributions.** In this paper, we propose a novel accelerated variance reduction type method, namely the **va**riance-**r**educed **a**ccelerated **g**radient (Varag ) method, to solve smooth finite-sum optimization problems given in the form of (1.1). Table 2 summarizes the main convergence results achieved by our Varag algorithm.

Table 2: Summary of the main convergence results for Varag

| Problem | Relations of $m$, $1/\epsilon$ and $L/\mu$ | Unified results |
|---|---|---|
| smooth optimization problems (1.1) with or without strong convexity | $m\geq\frac{D_0}{\epsilon}$ [2] or $m\geq\frac{3L}{4\mu}$ | $\mathcal{O}\left\{m\log\frac{1}{\epsilon}\right\}$ |
| | $m<\frac{D_0}{\epsilon}\leq\frac{3L}{4\mu}$ | $\mathcal{O}\left\{m\log m+\sqrt{\frac{mL}{\epsilon}}\right\}$ |
| | $m<\frac{3L}{4\mu}\leq\frac{D_0}{\epsilon}$ | $\mathcal{O}\left\{m\log m+\sqrt{\frac{mL}{\mu}}\log\frac{D_0/\epsilon}{3L/4\mu}\right\}$ [3] |

Firstly, for smooth convex finite-sum optimization, our proposed method exploits a direct acceleration scheme instead of employing any perturbation or restarting techniques to obtain desired optimal convergence results. As shown in the first two rows of Table 2, Varag achieves the optimal rate of convergence if the number of component functions $m$ is relatively small and/or the required accuracy is high, while it exhibits a fast linear rate of convergence when the number of component functions $m$ is relatively large and/or the required accuracy is low, without requiring any strong convexity assumptions. To the best of our knowledge, this is the first time that these complexity bounds have been obtained through a direct acceleration scheme for smooth convex finite-sum optimization in the literature. In comparison with existing methods using perturbation techniques, Varag does not need to know the target accuracy or the diameter of the feasible region a priori, and thus can be used to solve a much wider class of smooth convex problems, e.g., those with unbounded feasible sets.

Secondly, we equip Varag with a unified step-size policy for smooth convex optimization no matter (1.1) is strongly convex or not, i.e., the strongly convex modulus $\mu\geq 0$. With this step-size policy, Varag can adjust to different classes of problems to achieve the best convergence results, without knowing the target accuracy and/or fixing the number of epochs. In particular, as shown in the last column of Table 2, when $\mu$ is relatively large, Varag achieves the well-known optimal linear rate of convergence. If $\mu$ is relatively small, e.g., $\mu<\epsilon$, it obtains the accelerated convergence rate that is independent of the condition number $L/\mu$. Therefore, Varag is robust against ill-conditioning of problem (1.1). Moreover, our assumptions on the objective function is more general comparing to those used by other RIG methods, such as RPDG and Katyusha. Specifically, Varag does not require to keep a strongly convex regularization term in the projection, and so we can assume that the strong convexity is associated with the smooth function $f$ instead of the simple proximal function $h(\cdot)$. Some other advantages of Varag over existing accelerated SVRG methods, e.g., Katyusha,

include that it only requires the solution of one, rather than two, subproblems, and that it can allow the application of non-Euclidean Bregman distance for solving all different classes of problems.

Finally, we extend Varag to solve two more general classes of finite-sum optimization problems. We demonstrate that Varag is the first randomized method that achieves the accelerated linear rate of convergence when solving the class of problems that satisfies a certain error-bound condition rather than strong convexity. We then show that Varag can also be applied to solve stochastic smooth finite-sum optimization problems resulting in a sublinear rate of convergence.

This paper is organized as follows. In Section 2, we present our proposed algorithm Varag and its convergence results for solving (1.1) under different problem settings. In Section 3 we provide extensive experimental results to demonstrate the advantages of Varag over several state-of-the-art methods for solving some well-known ML models, e.g., logistic regression, Lasso, etc. We defer the proofs of the main results in Appendix A.

**Notation and terminology.** We use $\|\cdot\|$ to denote a general norm in $\mathbb{R}^n$ without specific mention, and $\|\cdot\|_*$ to denote the conjugate norm of $\|\cdot\|$. For any $p \geq 1$, $\|\cdot\|_p$ denotes the standard $p$-norm in $\mathbb{R}^n$, i.e., $\|x\|_p^p = \sum_{i=1}^n |x_i|^p$, for any $x \in \mathbb{R}^n$. For a given strongly convex function $w : X \to \mathbb{R}$ with modulus 1 w.r.t. an arbitrary norm $\|\cdot\|$, we define a *prox-function* associated with $w$ as

$$V(x^0, x) \equiv V_w(x^0, x) := w(x) - \left[w(x^0) + \langle w'(x^0), x - x^0 \rangle\right], \qquad (1.5)$$

where $w'(x^0) \in \partial w(x^0)$ is any subgradient of $w$ at $x^0$. By the strong convexity of $w$, we have

$$V(x^0, x) \geq \tfrac{1}{2}\|x - x^0\|^2, \quad \forall x, x^0 \in X. \qquad (1.6)$$

Notice that $V(\cdot, \cdot)$ described above is different from the standard definition for Bregman distance [6, 3, 4, 15, 7] in the sense that $w$ is not necessarily differentiable. Throughout this paper, we assume that the prox-mapping associated with $X$ and $h$, given by

$$\text{argmin}_{x \in X} \left\{\gamma[\langle g, x \rangle + h(x) + \mu V(\underline{x}_0, x)] + V(x_0, x)\right\}, \qquad (1.7)$$

can be easily computed for any $\underline{x}_0, x_0 \in X, g \in \mathbb{R}^n, \mu \geq 0, \gamma > 0$. We denote logarithm with base 2 as log. For any real number $r$, $\lceil r \rceil$ and $\lfloor r \rfloor$ denote the ceiling and floor of $r$.

## 2 Algorithms and main results

This section contains two subsections. We first present in Subsection 2.1 a unified optimal Varag for solving the finite-sum problem given in (1.1) as well as its optimal convergence results. Subsection 2.2 is devoted to the discussion of several extensions of Varag . Throughout this section, we assume that each component function $f_i$ is smooth with $L_i$-Lipschitz continuous gradients over $X$, i.e., (1.2) holds for all component functions. Moreover, we assume that the objective function $\psi(x)$ is possibly strongly convex, in particular, for $f(x) = \frac{1}{m}\sum_{i=1}^m f_i(x), \exists \mu \geq 0$ s.t.

$$f(y) \geq f(x) + \langle \nabla f(x), y - x \rangle + \mu V(x, y), \forall x, y \in X. \qquad (2.1)$$

Note that we assume the strong convexity of $\psi$ comes from $f$, and the simple function $h$ is not necessarily strongly convex. Clearly the strong convexity of $h$, if any, can be shifted to $f$ since $h$ is assumed to be simple and its structural information is transparent to us. Also observe that (2.1) is defined based on a generalized Bregman distance, and together with (1.6) they imply the standard definition of strong convexity w.r.t. Euclidean norm.

### 2.1 Varag **for convex finite-sum optimization**

The basic scheme of Varag is formally described in Algorithm 1. In each epoch (or outer loop), it first computes the full gradient $\nabla f(\tilde{x})$ at the point $\tilde{x}$ (cf. Line 3), which will then be repeatedly used to define a gradient estimator $G_t$ at each iteration of the inner loop (cf. Line 8). This is the well-known variance reduction technique employed by many algorithms (e.g., [14, 27, 1, 13]). The inner loop has a similar algorithmic scheme to the accelerated stochastic approximation algorithm [17, 11, 12] with a constant step-size policy. Indeed, the parameters used in the inner loop, i.e., $\{\gamma_s\}, \{\alpha_s\}$, and $\{p_s\}$, only depend on the index of epoch $s$. Each iteration of the inner loop requires the gradient information of only one randomly selected component function $f_{i_t}$, and maintains three primal sequences, $\{\underline{x}_t\}, \{x_t\}$ and $\{\bar{x}_t\}$, which play important role in the acceleration scheme.

---

**Algorithm 1** The variance-reduced accelerated gradient (Varag ) method

---
**Input:** $x^0 \in X, \{T_s\}, \{\gamma_s\}, \{\alpha_s\}, \{p_s\}, \{\theta_t\}$, and a probability distribution $Q = \{q_1, \dots, q_m\}$ on $\{1, \dots, m\}$.
1: Set $\tilde{x}^0 = x^0$.
2: **for** $s = 1, 2, \dots$ **do**
3:     Set $\tilde{x} = \tilde{x}^{s-1}$ and $\tilde{g} = \nabla f(\tilde{x})$.
4:     Set $x_0 = x^{s-1}, \bar{x}_0 = \tilde{x}$ and $T = T_s$.
5:     **for** $t = 1, 2, \dots, T$ **do**
6:         Pick $i_t \in \{1, \dots, m\}$ randomly according to $Q$.
7:         $\underline{x}_t = [(1 + \mu\gamma_s)(1 - \alpha_s - p_s)\bar{x}_{t-1} + \alpha_s x_{t-1} + (1 + \mu\gamma_s)p_s\tilde{x}] / [1 + \mu\gamma_s(1 - \alpha_s)]$.
8:         $G_t = (\nabla f_{i_t}(\underline{x}_t) - \nabla f_{i_t}(\tilde{x}))/(q_{i_t}m) + \tilde{g}$.
9:         $x_t = \arg\min_{x \in X} \{\gamma_s [\langle G_t, x\rangle + h(x) + \mu V(\underline{x}_t, x)] + V(x_{t-1}, x)\}$.
10:         $\bar{x}_t = (1 - \alpha_s - p_s)\bar{x}_{t-1} + \alpha_s x_t + p_s\tilde{x}$.
11:     **end for**
12:     Set $x^s = x_T$ and $\tilde{x}^s = \sum_{t=1}^{T}(\theta_t\bar{x}_t)/\sum_{t=1}^{T}\theta_t$.
13: **end for**

---

Note that Varag is closely related to stochastic mirror descent method [22, 23] and SVRG[14, 27]. By setting $\alpha_s = 1$ and $p_s = 0$, Algorithm 1 simply combines the variance reduction technique with stochastic mirror descent. In this case, the algorithm only maintains one primal sequence $\{x_t\}$ and possesses the non-accelerated rate of convergence $\mathcal{O}\{(m + L/\mu)\log(1/\epsilon)\}$ for solving (1.1). Interestingly, if we use Euclidean distance instead of prox-function $V(\cdot, \cdot)$ to update $x_t$ and set $X = \mathbb{R}^n$, Algorithm 1 will further reduce to prox-SVRG proposed in [27].

It is also interesting to observe the difference between Varag and Katyusha [1] because both are accelerated variance reduction methods. Firstly, while Katyusha needs to assume that the strongly convex term is specified as in the form of a simple proximal function, e.g., $\ell_1/\ell_2$-regularizer, Varag assumes that $f$ is possibly strongly convex, which solves an open issue of the existing accelerated RIG methods pointed out by [25]. Therefore, the momentum steps in Lines 7 and 10 are different from Katyusha. Secondly, Varag has a less computationally expensive algorithmic scheme. Particularly, Varag only needs to solve one proximal mapping (cf. Line 9) per iteration even if $f$ is strongly convex, while Katyusha requires to solve two proximal mappings per iteration. Thirdly, Varag incorporates a prox-function $V$ defined in (1.5) rather than the Euclidean distance in the proximal mapping to update $x_t$. This allows the algorithm to take advantage of the geometry of the constraint set $X$ when performing projections. However, Katyusha cannot be fully adapted to the non-Euclidean setting because its second proximal mapping must be defined using the Euclidean distance regardless the strong convexity of $\psi$. Finally, we will show in this section that Varag can achieve a much better rate of convergence than Katyusha for smooth convex finite-sum optimization by using a novel approach to specify step-size and to schedule epoch length.

We first discuss the case when $f$ is not necessarily strongly convex, i.e., $\mu = 0$ in (2.1). In Theorem 1, we suggest one way to specify the algorithmic parameters, including $\{q_i\}, \{\theta_t\}, \{\alpha_s\}, \{\gamma_s\}, \{p_s\}$ and $\{T_s\}$, for Varag to solve smooth convex problems given in the form of (1.1), and discuss its convergence properties of the resulting algorithm. We defer the proof of this result in Appendix A.1.

**Theorem 1 (Smooth finite-sum optimization)** *Suppose that the probabilities $q_i$'s are set to $L_i/\sum_{i=1}^{m} L_i$ for $i = 1, \dots, m$, and weights $\{\theta_t\}$ are set as*

$$\theta_t = \begin{cases} \frac{\gamma_s}{\alpha_s}(\alpha_s + p_s) & 1 \leq t \leq T_s - 1 \\ \frac{\gamma_s}{\alpha_s} & t = T_s. \end{cases} \quad (2.2)$$

*Moreover, let us denote $s_0 := \lfloor \log m \rfloor + 1$ and set parameters $\{T_s\}, \{\gamma_s\}$ and $\{p_s\}$ as*

$$T_s = \begin{cases} 2^{s-1}, & s \leq s_0 \\ T_{s_0}, & s > s_0 \end{cases}, \ \gamma_s = \frac{1}{3L\alpha_s}, \ and \ p_s = \frac{1}{2}, \ with \quad (2.3)$$

$$\alpha_s = \begin{cases} \frac{1}{2}, & s \leq s_0 \\ \frac{2}{s-s_0+4}, & s > s_0 \end{cases}. \quad (2.4)$$

*Then the total number of gradient evaluations of $f_i$ performed by Algorithm 1 to find a stochastic $\epsilon$-solution of (1.1), i.e., a point $\bar{x} \in X$ s.t. $\mathbb{E}[\psi(\bar{x}) - \psi^*] \leq \epsilon$, can be bounded by*

$$\bar{N} := \begin{cases} \mathcal{O}\left\{m \log \frac{D_0}{\epsilon}\right\}, & m \geq D_0/\epsilon, \\ \mathcal{O}\left\{m \log m + \sqrt{\frac{mD_0}{\epsilon}}\right\}, & m < D_0/\epsilon, \end{cases} \tag{2.5}$$

*where $D_0$ is defined as*

$$D_0 := 2[\psi(x^0) - \psi(x^*)] + 3LV(x^0, x^*). \tag{2.6}$$

We now make a few observations regarding the results obtained in Theorem 1. Firstly, as mentioned earlier, whenever the required accuracy $\epsilon$ is low and/or the number of components $m$ is large, Varag can achieve a fast linear rate of convergence even under the assumption that the objective function is not strongly convex. Otherwise, Varag achieves an optimal sublinear rate of convergence with complexity bounded by $\mathcal{O}\{\sqrt{mD_0/\epsilon} + m \log m\}$. Secondly, whenever $\sqrt{mD_0/\epsilon}$ is dominating in the second case of (2.5), Varag can save up to $\mathcal{O}(\sqrt{m})$ gradient evaluations of $f_i$ than the optimal deterministic first-order methods for solving (1.1). To the best our knowledge, Varag is the first accelerated RIG in the literature to obtain such convergence results by directly solving (1.1). Other existing accelerated RIG methods, such as RPDG[18] and Katyusha[1], require the application of perturbation and restarting techniques to obtain such convergence results. Thirdly, Varag also supports mini-batch approach where the component function $f_i$ is associated with a mini-batch of data samples instead of a single data sample. In a more general case, for a given mini-batch size $b$, we assume that the component functions can be split into subsets where each subset contains exactly $b$ number of component functions. Therefore, one can replace Line 8 in Algorithm 1 by $G_t = \frac{1}{b}\sum_{i_t \in S_b}(\nabla f_{i_t}(\underline{x}_t) - \nabla f_{i_t}(\tilde{x}))/(q_{i_t}m) + \tilde{g}$ with $S_b$ being the selected subset and $|S_b| = b$ and adjust the appropriate parameters to obtain the mini-batch version of Varag . The mini-batch Varag can obtain parallel linear speedup of factor $b$ whenever the mini-batch size $b \leq \sqrt{m}$.

Next we consider the case when $f$ is possibly strongly convex, including the situation when the problem is almost not strongly convex, i.e., $\mu \approx 0$. In the latter case, the term $\sqrt{mL/\mu}\log(1/\epsilon)$ will be dominating in the complexity of existing accelerated RIG methods (e.g., [18, 19, 1, 20]) and will tend to $\infty$ as $\mu$ decreases. Therefore, these complexity bounds are significantly worse than (2.5) obtained by simply treating (1.1) as smooth convex problems. Moreover, $\mu \approx 0$ is very common in ML applications. In Theorem 2, we provide a unified step-size policy which allows Varag to achieve optimal rate of convergence for finite-sum optimization in (1.1) regardless of its strong convexity, and hence it can achieve stronger rate of convergence than existing accelerated RIG methods if the condition number $L/\mu$ is very large. The proof of this result can be found in Appendix A.2.

**Theorem 2 (A unified result for convex finite-sum optimization)** *Suppose that the probabilities $q_i$'s are set to $L_i/\sum_{i=1}^m L_i$ for $i = 1, \ldots, m$. Moreover, let us denote $s_0 := \lfloor \log m \rfloor + 1$ and assume that the weights $\{\theta_t\}$ are set to (2.2) if $1 \leq s \leq s_0$ or $s_0 < s \leq s_0 + \sqrt{\frac{12L}{m\mu}} - 4$, $m < \frac{3L}{4\mu}$. Otherwise, they are set to*

$$\theta_t = \begin{cases} \Gamma_{t-1} - (1 - \alpha_s - p_s)\Gamma_t, & 1 \leq t \leq T_s - 1, \\ \Gamma_{t-1}, & t = T_s, \end{cases} \tag{2.7}$$

*where $\Gamma_t = (1 + \mu\gamma_s)^t$. If the parameters $\{T_s\}$, $\{\gamma_s\}$ and $\{p_s\}$ set to (2.3) with*

$$\alpha_s = \begin{cases} \frac{1}{2}, & s \leq s_0, \\ \max\left\{\frac{2}{s-s_0+4}, \min\{\sqrt{\frac{m\mu}{3L}}, \frac{1}{2}\}\right\}, & s > s_0, \end{cases} \tag{2.8}$$

*then the total number of gradient evaluations of $f_i$ performed by Algorithm 1 to find a stochastic $\epsilon$-solution of (1.1) can be bounded by*

$$\bar{N} := \begin{cases} \mathcal{O}\left\{m \log \frac{D_0}{\epsilon}\right\}, & m \geq \frac{D_0}{\epsilon} \text{ or } m \geq \frac{3L}{4\mu}, \\ \mathcal{O}\left\{m \log m + \sqrt{\frac{mD_0}{\epsilon}}\right\}, & m < \frac{D_0}{\epsilon} \leq \frac{3L}{4\mu}, \\ \mathcal{O}\left\{m \log m + \sqrt{\frac{mL}{\mu}} \log \frac{D_0/\epsilon}{3L/4\mu}\right\}, & m < \frac{3L}{4\mu} \leq \frac{D_0}{\epsilon}. \end{cases} \tag{2.9}$$

*where $D_0$ is defined as in (2.6).*

Observe that the complexity bound (2.9) is a unified convergence result for Varag to solve deterministic smooth convex finite-sum optimization problems (1.1). When the strong convex modulus $\mu$ of the objective function is large enough, i.e., $3L/\mu < D_0/\epsilon$, Varag exhibits an optimal linear rate of convergence since the third case of (2.9) matches the lower bound (1.4) for RIG methods. If $\mu$ is relatively small, Varag treats the finite-sum problem (1.1) as a smooth problem without strong convexity, which leads to the same complexity bounds as in Theorem 1. It should be pointed out that the parameter setting proposed in Theorem 2 does not require the values of $\epsilon$ and $D_0$ given a priori.

## 2.2 Generalization of Varag

In this subsection, we extend Varag to solve two general classes of finite-sum optimization problems as well as establishing its convergence properties for these problems.

**Finite-sum problems under error bound condition.** We investigate a class of weakly strongly convex problems, i.e., $\psi(x)$ is smooth convex and satisfies the error bound condition given by

$$V(x, X^*) \leq \tfrac{1}{\bar{\mu}}(\psi(x) - \psi^*), \ \forall x \in X, \tag{2.10}$$

where $X^*$ denotes the set of optimal solutions of (1.1). Many optimization problems satisfy (2.10), for instance, linear systems, quadratic programs, linear matrix inequalities and composite problems (outer: strongly convex, inner: polyhedron functions), see [8] and Section 6 of [21] for more examples. Although these problems are not strongly convex, by properly restarting Varag we can solve them with an accelerated linear rate of convergence, the best-known complexity result to solve this class of problems so far. We formally present the result in Theorem 3, whose proof is given in Appendix A.3.

**Theorem 3 (Convex finite-sum optimization under error bound)** *Assume that the probabilities $q_i$'s are set to $L_i/\sum_{i=1}^m L_i$ for $i = 1, \ldots, m$, and $\theta_t$ are defined as (2.2). Moreover, let us set parameters $\{\gamma_s\}$, $\{p_s\}$ and $\{\alpha_s\}$ as in (2.3) and (2.4) with $\{T_s\}$ being set as*

$$T_s = \begin{cases} T_1 2^{s-1}, & s \leq 4 \\ 8T_1, & s > 4 \end{cases}, \tag{2.11}$$

*where $T_1 = \min\{m, \frac{L}{\bar{\mu}}\}$. Then under condition (2.10), for any $x^* \in X^*$, $s = 4 + 4\sqrt{\frac{L}{\bar{\mu}m}}$,*

$$\mathbb{E}[\psi(\tilde{x}^s) - \psi(x^*)] \leq \tfrac{5}{16}[\psi(x^0) - \psi(x^*)]. \tag{2.12}$$

*Moreover, if we restart Varag every time it runs $s$ iterations for $k = \log \frac{\psi(x^0) - \psi(x^*)}{\epsilon}$ times, the total number of gradient evaluations of $f_i$ to find a stochastic $\epsilon$-solution of (1.1) can be bounded by*

$$\bar{N} := k(\textstyle\sum_s (m + T_s)) = \mathcal{O}\left\{\left(m + \sqrt{\frac{mL}{\bar{\mu}}}\right) \log \frac{\psi(x^0) - \psi(x^*)}{\epsilon}\right\}. \tag{2.13}$$

**Remark 1** *Note that Varag can also be extended to obtain an unified result as shown in Theorem 2 for solving finite-sum problems under error bound condition. In particular, if the condition number is very large, i.e., $s = \mathcal{O}\{L/(\bar{\mu}m)\} \approx \infty$, Varag will never be restarted, and the resulting complexity bounds will reduce to the case for solving smooth convex problems provided in Theorem 1.*

**Stochastic finite-sum optimization.** We now consider stochastic smooth convex finite-sum optimization and online learning problems defined as in (1.3), where only noisy gradient information of $f_i$ can be accessed via a SFO oracle. In particular, for any $x \in X$, the SFO oracle outputs a vector $G_i(x, \xi_j)$ such that

$$\mathbb{E}_{\xi_j}[G_i(x, \xi_j)] = \nabla f_i(x), \ i = 1, \ldots, m, \tag{2.14}$$

$$\mathbb{E}_{\xi_j}[\|G_i(x, \xi_j) - \nabla f_i(x)\|_*^2] \leq \sigma^2, \ i = 1, \ldots, m. \tag{2.15}$$

We present the variant of Varag for stochastic finite-sum optimization in Algorithm 2 as well as its convergence results in Theorem 4, whose proof can be found in Appendix B.

**Theorem 4 (Stochastic smooth finite-sum optimization)** *Assume that $\theta_t$ are defined as in (2.2), $C := \sum_{i=1}^m \frac{1}{q_i m^2}$ and the probabilities $q_i$'s are set to $L_i/\sum_{i=1}^m L_i$ for $i = 1, \ldots, m$. Moreover, let us*

---

**Algorithm 2** Stochastic accelerated variance-reduced stochastic gradient descent (Stochastic Varag )

---

This algorithm is the same as Algorithm 1 except that for given batch-size parameters $B_s$ and $b_s$, Line 3 is replaced by $\tilde{x} = \tilde{x}^{s-1}$ and

$$\tilde{g} = \frac{1}{m}\sum_{i=1}^{m}\left\{G_i(\tilde{x}) := \frac{1}{B_s}\sum_{j=1}^{B_s}G_i(\tilde{x}, \xi_j^s)\right\}, \tag{2.16}$$

and Line 8 is replaced by

$$G_t = \frac{1}{q_{i_t}mb_s}\sum_{k=1}^{b_s}\left(G_{i_t}(\underline{x}_t, \xi_k^s) - G_{i_t}(\tilde{x})\right) + \tilde{g}. \tag{2.17}$$

---

*denote $s_0 := \lfloor \log m \rfloor + 1$ and set $T_s$, $\alpha_s$, $\gamma_s$ and $p_s$ as in (2.3) and (2.4). Then the number of calls to the SFO oracle required by Algorithm 2 to find a stochastic $\epsilon$-solution of* (1.1) *can be bounded by*

$$N_{\text{SFO}} = \sum_s(mB_s + T_sb_s) = \begin{cases} \mathcal{O}\left\{\frac{mC\sigma^2}{L\epsilon}\right\}, & m \geq D_0/\epsilon, \\ \mathcal{O}\left\{\frac{C\sigma^2 D_0}{L\epsilon^2}\right\}, & m < D_0/\epsilon, \end{cases} \tag{2.18}$$

*where $D_0$ is given in* (2.6).

**Remark 2** *Note that the constant $C$ in* (2.18) *can be easily upper bounded by $\frac{L}{\min\{L_i\}}$, and $C = 1$ if $L_i = L, \forall i$. To the best of our knowledge, among a few existing RIG methods that can be applied to solve the class of stochastic finite-sum problems, Varag is the first to achieve such complexity results as in* (2.18) *for smooth convex problems. RGEM[19] obtains nearly-optimal rate of convergence for strongly convex case, but cannot solve stochastic smooth problems directly, and [16] required a specific initial point, i.e., an exact solution to a proximal mapping depending on the variance $\sigma^2$, to achieve $\mathcal{O}\left\{m \log m + \sigma^2/\epsilon^2\right\}$ rate of convergence for smooth convex problems.*

# 3 Numerical experiments

In this section, we demonstrate the advantages of our proposed algorithm, Varag over several state-of-the-art algorithms, e.g., SVRG++ [2] and Katyusha [1], etc., via solving several well-known machine learning models. For all experiments, we use public real datasets downloaded from UCI Machine Learning Repository [10] and uniform sampling strategy to select $f_i$. Indeed, the theoretical suggesting sampling distribution should be non-uniform, i.e., $q_i = L_i/\sum_{i=1}^{m}L_i$, which results in the optimal constant $L$ appearing in the convergence results. However, a uniform sampling strategy will only lead to a constant factor slightly larger than $L = \frac{1}{m}\sum_{i=1}^{m}L_i$. Moreover, it is computationally efficient to estimate $L_i$ by performing maximum singular value decomposition of the Hessian since only a rough estimation suffices.

**Unconstrained smooth convex problems.** We first investigate unconstrained logistic models which cannot be solved via the perturbation approach due to the unboundedness of the feasible set. More specifically, we applied Varag , SVRG++ and Katyusha[ns] to solve a logistic regression problem,

$$\min_{x\in\mathbb{R}^n}\{\psi(x) := \frac{1}{m}\sum_{i=1}^{m}f_i(x)\} \text{ where } f_i(x) := \log(1 + \exp(-b_ia_i^Tx))\}. \tag{3.1}$$

Here $(a_i, b_i) \in \mathbb{R}^n \times \{-1, 1\}$ is a training data point and $m$ is the sample size, and hence $f_i$ now corresponds to the loss generated by a single training data. As we can see from Figure 1, Varag converges much faster than SVRG++ and Katyusha in terms of training loss.

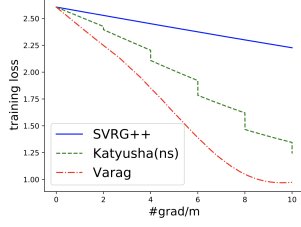
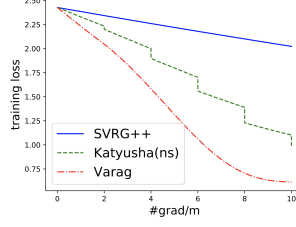

Diabetes ($m = 1151$),
unconstrained logistic

Breast Cancer Wisconsin ($m = 683$),
unconstrained logistic

Figure 1: The algorithmic parameters for SVRG++ and Katyusha[ns] are set according to [2] and [1], respectively, and those for Varag are set as in Theorem 1.

**Strongly convex loss with simple convex regularizer.** We now study the class of Lasso regression problems with $\lambda$ as the regularizer coefficient, given in the following form

$$\min_{x \in \mathbb{R}^n} \{\psi(x) := \frac{1}{m}\sum_{i=1}^{m} f_i(x) + h(x)\} \text{ where } f_i(x) := \frac{1}{2}(a_i^T x - b_i)^2, h(x) := \lambda \|x\|_1. \quad (3.2)$$

Due to the assumption SVRG++ and Katyusha enforced on the objective function that the strong convexity can only be associated with the regularizer, these methods always view Lasso as smooth problems [25], while Varag can treat Lasso as strongly convex problems. As can be seen from Figure 2, Varag outperforms SVRG++ and Katyusha[ns] in terms of training loss.

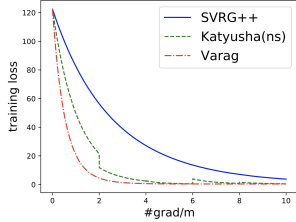
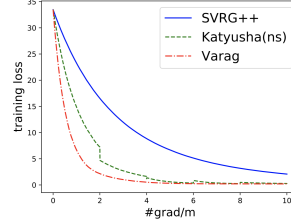

Diabetes ($m = 1151$),
Lasso $\lambda = 0.001$

Breast Cancer Wisconsin ($m = 683$),
Lasso $\lambda = 0.001$

Figure 2: The algorithmic parameters for SVRG++ and Katyusha[ns] are set according to [2] and [1], respectively, and those for Varag are set as in Theorem 2.

**Weakly strongly convex problems satisfying error bound condition.** Let us consider a special class of finite-sum convex quadratic problems given in the following form

$$\min_{x \in \mathbb{R}^n} \{\psi(x) := \frac{1}{m}\sum_{i=1}^{m} f_i(x)\} \text{ where } f_i(x) := \frac{1}{2}x^T Q_i x + q_i^T x. \quad (3.3)$$

Here $q_i = -Q_i x_s$ and $x_s$ is a solution to the symmetric linear system $Q_i x + q_i = 0$ with $Q_i \succeq 0$. [8][Section 6] and [21][Section 6.1] proved that (3.3) belongs to the class of weakly strongly convex problems satisfying error bound condition (2.10). For a given solution $x_s$, we use the following real datasets to generate $Q_i$ and $q_i$. We then compare the performance of Varag with fast gradient method (FGM) proposed in [21]. As shown in Figure 3, Varag outperforms FGM for all cases. And as the number of component functions $m$ increases, Varag demonstrates more advantages over FGM. These numerical results are consistent with the theoretical complexity bound (2.13) suggesting that Varag can save up to $\mathcal{O}\{\sqrt{m}\}$ number of gradient computations than deterministic algorithms, e.g., FGM.

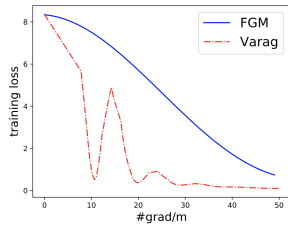
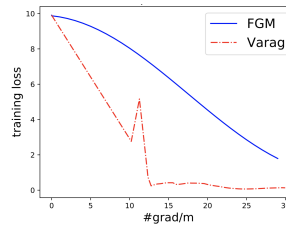

Diabetes ($m = 1151$)

Parkinsons Telemonitoring ($m = 5875$)

Figure 3: The algorithmic parameters for FGM and Varag are set according to [21] and Theorem 3, respectively.

More numerical experiment results on another problem case, strongly convex problems with small strongly convex modulus, can be found in Appendix C.

## Footnotes

[1]These complexity bounds are obtained via indirect approaches, i.e., by adding strongly convex perturbation.

[2]$D_0=2[\psi(x^0)-\psi(x^*)]+3LV(x^0,x^*)$ where $x^0$ is the initial point, $x^*$ is the optimal solution of (1.1) and $V$ is defined in (1.5).

[3]Note that this term is less than $\mathcal{O}\{\sqrt{\frac{mL}{\mu}}\log\frac{1}{\epsilon}\}$.

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
