[Supplementary Material]

## A Convergence analysis of Varag for deterministic finite-sum optimization

Our main goal in this section is to establish the convergence results stated in Theorems 1 and 2 for the Varag method applied to the finite-sum optimization problem in (1.1).

Before proving Theorem 1 and 2, we first need to present some basic properties for smooth convex functions and then provide some important technical results.

**Lemma 1** *If $f : X \to \mathbb{R}$ has Lipschitz continuous gradients with Lipschitz constant $L$, then*

$$\tfrac{1}{2L}\|\nabla f(x) - \nabla f(z)\|_*^2 \le f(x) - f(z) - \langle \nabla f(z), x - z \rangle \ \ \forall x, z \in X.$$

*Proof:* Denote $\phi(x) = f(x) - f(z) - \langle \nabla f(z), x - z \rangle$. Clearly $\phi$ also has $L$-Lipschitz continuous gradients. It is easy to check that $\nabla \phi(z) = 0$, and hence that $\min_x \phi(x) = \phi(z) = 0$, which implies

$$\phi(z) \le \phi(x - \tfrac{1}{L}\nabla \phi(x))$$

$$= \phi(x) + \int_0^1 \langle \nabla \phi \left( x - \tfrac{\tau}{L}\nabla \phi(x) \right), -\tfrac{1}{L}\nabla \phi(x) \rangle d\tau$$

$$= \phi(x) + \langle \nabla \phi(x), -\tfrac{1}{L}\nabla \phi(x) \rangle + \int_0^1 \langle \nabla \phi \left( x - \tfrac{\tau}{L}\nabla \phi(x) \right) - \nabla \phi(x), -\tfrac{1}{L}\nabla \phi(x) \rangle d\tau$$

$$\le \phi(x) - \tfrac{1}{L}\|\nabla \phi(x)\|_*^2 + \int_0^1 L\|\tfrac{\tau}{L}\nabla \phi(x)\|_* \|\tfrac{1}{L}\nabla \phi(x)\|_* d\tau$$

$$= \phi(x) - \tfrac{1}{2L}\|\nabla \phi(x)\|_*^2.$$

Therefore, we have $\tfrac{1}{2L}\|\nabla \phi(x)\|_*^2 \le \phi(x) - \phi(z) = \phi(x)$, and the result follows immediately from this relation. $\qquad\square$

The following result follows as a consequence of Lemma 1.

**Lemma 2** *Let $x^*$ be an optimal solution of* (1.1)*. Then we have*

$$\tfrac{1}{m}\sum_{i=1}^m \tfrac{1}{mq_i}\|\nabla f_i(x) - \nabla f_i(x^*)\|_*^2 \le 2L_Q\left[\psi(x) - \psi(x^*)\right], \ \forall x \in X, \tag{A.1}$$

*where*

$$L_Q = \tfrac{1}{m} \max_{i=1,\dots,m} \tfrac{L_i}{q_i}. \tag{A.2}$$

*Proof:* By Lemma 1 (with $f = f_i$), we have

$$\|\nabla f_i(x) - \nabla f_i(x^*)\|_*^2 \le 2L_i\left[f_i(x) - f_i(x^*) - \langle \nabla f_i(x^*), x - x^* \rangle\right].$$

Dividing this inequality by $1/(m^2 q_i)$, and summing over $i = 1, \dots, m$, we obtain

$$\tfrac{1}{m}\sum_{i=1}^m \tfrac{1}{mq_i}\|\nabla f_i(x) - \nabla f_i(x^*)\|_*^2 \le 2L_Q\left[f(x) - f(x^*) - \langle \nabla f(x^*), x - x^* \rangle\right]. \tag{A.3}$$

By the optimality of $x^*$, we have $\langle \nabla f(x^*) + h'(x^*), x - x^* \rangle \ge 0$ for any $x \in X$, which in view of the convexity of $h$, implies that $\langle \nabla f(x^*), x - x^* \rangle \ge h(x^*) - h(x)$ for any $x \in X$. The result then follows by combining the previous two conclusions. $\qquad\square$

In the sequel, let us define some important notations that help us to simplify the convergence analysis of Varag .

$$l_f(z, x) := f(z) + \langle \nabla f(z), x - z \rangle, \tag{A.4}$$

$$\delta_t := G_t - \nabla f(\underline{x}_t), \tag{A.5}$$

$$x_{t-1}^+ := \tfrac{1}{1+\mu\gamma_s}\left(x_{t-1} + \mu\gamma_s \underline{x}_t\right), \tag{A.6}$$

where $G_t$, $\underline{x}_t$ and $x_{t-1}$ are generated as in Algorithm 1. Lemma 3 below shows that $G_t$ is an unbiased estimator of $\nabla f(\underline{x}_t)$ and provides a tight upper bound for its variance.

**Lemma 3** *Conditionally on $x_1, \dots, x_{t-1}$,*

$$\mathbb{E}[\delta_t] = 0, \tag{A.7}$$

$$\mathbb{E}[\|\delta_t\|_*^2] \le 2L_Q[f(\tilde{x}) - f(\underline{x}_t) - \langle \nabla f(\underline{x}_t), \tilde{x} - \underline{x}_t \rangle]. \tag{A.8}$$

23   *Proof:* We take the expectation with respect to $i_t$ conditionally on $x_1, \ldots, x_t$, to obtain

$$\mathbb{E}\left[\tfrac{1}{mq_{i_t}}\nabla f_{i_t}(\underline{x}_t)\right] = \sum_{i=1}^{m}\tfrac{q_i}{mq_i}\nabla f_i(\underline{x}_t) = \sum_{i=1}^{m}\tfrac{1}{m}\nabla f_i(\underline{x}_t) = \nabla f(\underline{x}_t).$$

24   Similarly we have $\mathbb{E}\left[\tfrac{1}{mq_{i_t}}\nabla f_{i_t}(\tilde{x})\right] = \nabla f(\tilde{x})$. Therefore,

$$\mathbb{E}[G_t] = \mathbb{E}\left[\tfrac{1}{mq_{i_t}}\left(\nabla f_{i_t}(\underline{x}_t) - \nabla f_{i_t}(\tilde{x})\right) + \nabla f(\tilde{x})\right] = \nabla f(\underline{x}_t).$$

25   To bound the variance, we have

$$
\begin{aligned}
\mathbb{E}[\|\delta_t\|_*^2] &= \mathbb{E}[\|\tfrac{1}{mq_{i_t}}\left(\nabla f_{i_t}(\underline{x}_t) - \nabla f_{i_t}(\tilde{x})\right) + \nabla f(\tilde{x}) - \nabla f(\underline{x}_t)\|_*^2] \\
&= \mathbb{E}[\tfrac{1}{(mq_{i_t})^2}\|\nabla f_{i_t}(\underline{x}_t) - \nabla f_{i_t}(\tilde{x})\|_*^2] - \|\nabla f(\underline{x}_t) - \nabla f(\tilde{x})\|_*^2 \\
&\leq \mathbb{E}[\tfrac{1}{(mq_{i_t})^2}\|\nabla f_{i_t}(\underline{x}_t) - \nabla f_{i_t}(\tilde{x})\|_*^2]
\end{aligned}
$$

26   The above relation, in view of relation (A.3) (with $x$ and $x^*$ replaced by $\tilde{x}$ and $\underline{x}_t$), then implies (A.8).

27   $\hfill\square$

28   Using the definition of $x_{t-1}^+$ in (A.6), and the definitions of $\underline{x}_t$ and $\bar{x}_t$ in Algorithm 1 (see Line 6 and

29   9), we have

$$
\begin{aligned}
\bar{x}_t - \underline{x}_t &= (1 - \alpha_s - p_s)\bar{x}_{t-1} + \alpha_s x_t + p_s \tilde{x} - \underline{x}_t \\
&= \alpha_s x_t + \tfrac{1}{1+\mu\gamma_s}\left\{[1 + \mu\gamma_s(1-\alpha_s)]\underline{x}_t - \alpha_s x_{t-1}\right\} - \underline{x}_t \\
&= \alpha_s(x_t - x_{t-1}^+).
\end{aligned}
\tag{A.9}
$$

30   We characterize the solutions of the prox-mapping (1.6) (or Line 8 of Algorithm 1) in Lemma 4

31   below.

32   **Lemma 4 ([2, Lemma 2])** *Let the convex function $p : X \to \mathbb{R}$, the points $\tilde{x}, \tilde{y} \in X$ and the scalars*

33   $\mu_1, \mu_2 \geq 0$ *be given. Let $w : X \to \mathbb{R}$ be a convex function and $V(x^0, x)$ be defined in (1.4). If*

$$u^* \in \mathrm{Argmin}\{p(u) + \mu_1 V(\tilde{x}, u) + \mu_2 V(\tilde{y}, u) : u \in X\}, \tag{A.10}$$

34   *then for any $u \in X$, we have*

$$p(u^*) + \mu_1 V(\tilde{x}, u^*) + \mu_2 V(\tilde{y}, u^*) \leq p(u) + \mu_1 V(\tilde{x}, u) + \mu_2 V(\tilde{y}, u) - (\mu_1 + \mu_2)V(u^*, u).$$

35   The following result examines the optimality conditions associated with the definition of $x_t$ in Line 8

36   of Algorithm 1.

37   **Lemma 5** *For any $x \in X$, we have*

$$
\begin{aligned}
\gamma_s[l_f(\underline{x}_t, x_t) - l_f(\underline{x}_t, x) + h(x_t) - h(x)] \leq{}& \gamma_s \mu V(\underline{x}_t, x) + V(x_{t-1}, x) - (1 + \mu\gamma_s)V(x_t, x) \\
&- \tfrac{1+\mu\gamma_s}{2}\|x_t - x_{t-1}^+\|^2 - \gamma_s\langle\delta_t, x_t - x\rangle.
\end{aligned}
$$

38   *Proof:* It follows from Lemma 4 and the definition of $x_t$ in Algorithm 1 that

$$
\begin{aligned}
\gamma_s[\langle G_t, x_t - x\rangle + h(x_t) - h(x) + \mu V(\underline{x}_t, x_t)] + V(x_{t-1}, x_t) \\
\leq \gamma_s \mu V(\underline{x}_t, x) + V(x_{t-1}, x) - (1 + \mu\gamma_s)V(x_t, x).
\end{aligned}
$$

39   Also observe that

$$\langle G_t, x_t - x\rangle = \langle\nabla f(\underline{x}_t), x_t - x\rangle + \langle\delta_t, x_t - x\rangle = l_f(\underline{x}_t, x_t) - l_f(\underline{x}_t, x) + \langle\delta_t, x_t - x\rangle$$

40   and

$$
\begin{aligned}
\gamma_s \mu V(\underline{x}_t, x_t) + V(x_{t-1}, x_t) &\geq \tfrac{1}{2}\left(\mu\gamma_s\|x_t - \underline{x}_t\|^2 + \|x_t - x_{t-1}\|^2\right) \\
&\geq \tfrac{1+\mu\gamma_s}{2}\|x_t - x_{t-1}^+\|^2,
\end{aligned}
$$

41   where the last inequality follows from the definition of $x_{t-1}^+$ in (A.6) and the convexity of $\|\cdot\|$. The

42   result then follows by combining the above three relations. $\hfill\square$

43   We now show the possible progress made by each inner iteration of the Varag method.

44 **Lemma 6** *Assume that $\alpha_s \in [0,1]$, $p_s \in [0,1]$ and $\gamma_s > 0$ satisfy*

$$1 + \mu\gamma_s - L\alpha_s\gamma_s > 0, \tag{A.11}$$

$$p_s - \frac{L_Q\alpha_s\gamma_s}{1+\mu\gamma_s-L\alpha_s\gamma_s} \geq 0. \tag{A.12}$$

45 *Then, conditional on $x_1, \ldots, x_{t-1}$, we have*

$$\frac{\gamma_s}{\alpha_s}\mathbb{E}[\psi(\bar{x}_t) - \psi(x)] + (1+\mu\gamma_s)\mathbb{E}[V(x_t,x)]$$
$$\leq \frac{\gamma_s}{\alpha_s}(1-\alpha_s-p_s)[\psi(\bar{x}_{t-1}) - \psi(x)] + \frac{\gamma_s p_s}{\alpha_s}[\psi(\tilde{x}) - \psi(x)] + V(x_{t-1},x) \tag{A.13}$$

46 *for any $x \in X$.*

47 *Proof:* Note that by the smoothness of $f$, the definition of $\bar{x}_t$, and (A.9), we have

$$f(\bar{x}_t) \leq l_f(\underline{x}_t, \bar{x}_t) + \frac{L}{2}\|\bar{x}_t - \underline{x}_t\|^2$$
$$= (1-\alpha_s-p_s)l_f(\underline{x}_t, \bar{x}_{t-1}) + \alpha_s l_f(\underline{x}_t, x_t) + p_s l_f(\underline{x}_t, \tilde{x}) + \frac{L\alpha_s^2}{2}\|x_t - x_{t-1}^+\|^2.$$

48 The above inequality, in view of Lemma 5 and the (strong) convexity of $f$, then implies that

$$f(\bar{x}_t) \leq (1-\alpha_s-p_s)l_f(\underline{x}_t, \bar{x}_{t-1})$$
$$+ \alpha_s\left[l_f(\underline{x}_t, x) + h(x) - h(x_t) + \mu V(\underline{x}_t, x) + \frac{1}{\gamma_s}V(x_{t-1},x) - \frac{1+\mu\gamma_s}{\gamma_s}V(x_t,x)\right]$$
$$+ p_s l_f(\underline{x}_t, \tilde{x}) - \frac{\alpha_s}{2\gamma_s}(1+\mu\gamma_s-L\alpha_s\gamma_s)\|x_t - x_{t-1}^+\|^2 - \alpha_s\langle\delta_t, x_t - x\rangle$$
$$\leq (1-\alpha_s-p_s)f(\bar{x}_{t-1}) + \alpha_s\left[\psi(x) - h(x_t) + \frac{1}{\gamma_s}V(x_{t-1},x) - \frac{1+\mu\gamma_s}{\gamma_s}V(x_t,x)\right]$$
$$+ p_s l_f(\underline{x}_t, \tilde{x}) - \frac{\alpha_s}{2\gamma_s}(1+\mu\gamma_s-L\alpha_s\gamma_s)\|x_t - x_{t-1}^+\|^2$$
$$- \alpha_s\langle\delta_t, x_t - x_{t-1}^+\rangle - \alpha_s\langle\delta_t, x_{t-1}^+ - x\rangle$$
$$\leq (1-\alpha_s-p_s)f(\bar{x}_{t-1}) + \alpha_s\left[\psi(x) - h(x_t) + \frac{1}{\gamma_s}V(x_{t-1},x) - \frac{1+\mu\gamma_s}{\gamma_s}V(x_t,x)\right]$$
$$+ p_s l_f(\underline{x}_t, \tilde{x}) + \frac{\alpha_s\gamma_s\|\delta_t\|_*^2}{2(1+\mu\gamma_s-L\alpha_s\gamma_s)} - \alpha_s\langle\delta_t, x_{t-1}^+ - x\rangle, \tag{A.14}$$

49 where the last inequality follows from the fact that $b\langle u,v\rangle - a\|v\|^2/2 \leq b^2\|u\|^2/(2a), \forall a > 0$. Note
50 that by (A.7), (A.8), (A.12) and the convexity of $f$, we have, conditional on $x_1, \ldots, x_{t-1}$,

$$p_s l_f(\underline{x}_t, \tilde{x}) + \frac{\alpha_s\gamma_s\mathbb{E}[\|\delta_t\|_*^2]}{2(1+\mu\gamma_s-L\alpha_s\gamma_s)} - \alpha_s\mathbb{E}[\langle\delta_t, x_{t-1}^+ - x\rangle]$$
$$\leq p_s l_f(\underline{x}_t, \tilde{x}) + \frac{L_Q\alpha_s\gamma_s}{1+\mu\gamma_s-L\alpha_s\gamma_s}[f(\tilde{x}) - l_f(\underline{x}_t, \tilde{x})]$$
$$\leq \left(p_s - \frac{L_Q\alpha_s\gamma_s}{1+\mu\gamma_s-L\alpha_s\gamma_s}\right)l_f(\underline{x}_t, \tilde{x}) + \frac{L_Q\alpha_s\gamma_s}{1+\mu\gamma_s-L\alpha_s\gamma_s}f(\tilde{x}) \leq p_s f(\tilde{x}).$$

51 Moreover, by convexity of $h$, we have $h(\bar{x}_t) \leq (1-\alpha_s-p_s)h(\bar{x}_{t-1}) + \alpha_s h(x_t) + p_s h(\tilde{x})$. Summing
52 up the previous three conclusions, we obtain

$$\mathbb{E}[\psi(\bar{x}_t) + \frac{\alpha_s(1+\mu\gamma_s)}{\gamma_s}V(x_t,x)] \leq (1-\alpha_s-p_s)\psi(\bar{x}_{t-1}) + p_s\psi(\tilde{x}) + \alpha_s\psi(x) + \frac{\alpha_s}{\gamma_s}V(x_{t-1},x).$$

53 The result then follows by subtracting $\psi(x)$ from both sides of the above inequality. $\qquad\square$

## A.1 Smooth convex problems

55 In this subsection, we assume that $f$ is not necessarily strongly convex, i.e., $\mu = 0$ in (2.1). Lemma 7
56 below shows possible decrease of functional value in each epoch of Varag for solving these problems.

57 **Lemma 7** *Assume that for each epoch $s$, $s \geq 1$, the parameters $\alpha_s$, $\gamma_s$, $p_s$ and $T_s$ are chosen such*
58 *that (A.11)-(A.12) hold. Also, let us set $\theta_t$ to (2.2). Moreover, let us denote*

$$\mathcal{L}_s := \frac{\gamma_s}{\alpha_s} + (T_s-1)\frac{\gamma_s(\alpha_s+p_s)}{\alpha_s}, \quad \mathcal{R}_s := \frac{\gamma_s}{\alpha_s}(1-\alpha_s) + (T_s-1)\frac{\gamma_s p_s}{\alpha_s}, \tag{A.15}$$

59 *and assume that*

$$w_s := \mathcal{L}_s - \mathcal{R}_{s+1} \geq 0, \forall s \geq 1. \tag{A.16}$$

60 *Then we have*

$$\mathcal{L}_s \mathbb{E}[\psi(\tilde{x}^s) - \psi(x)] + (\sum_{j=1}^{s-1} w_j)\mathbb{E}[\psi(\bar{x}^s) - \psi(x)]$$
$$\leq \mathcal{R}_1 \mathbb{E}[\psi(\tilde{x}^0) - \psi(x)] + \mathbb{E}[V(x^0, x) - V(x^s, x)] \qquad (A.17)$$

61 *for any $x \in X$, where*

$$\bar{x}^s := (\sum_{j=1}^{s-1} w_j)\sum_{j=1}^{s-1}(w_j \tilde{x}^j). \qquad (A.18)$$

62 *Proof:* Using our assumptions on $\alpha_s$, $\gamma_s$ and $p_s$, and the fact that $\mu = 0$, we have

$$\frac{\gamma_s}{\alpha_s}\mathbb{E}[\psi(\bar{x}_t) - \psi(x)] \leq \frac{\gamma_s}{\alpha_s}(1 - \alpha_s - p_s)\mathbb{E}[\psi(\bar{x}_{t-1}) - \psi(x)]$$
$$+ \frac{\gamma_s p_s}{\alpha_s}\mathbb{E}[\psi(\tilde{x}) - \psi(x)] + \mathbb{E}[V(x_{t-1}, x) - V(x_t, x)].$$

63 Summing up these inequalities for $t = 1, \ldots, T_s$, using the definition of $\theta_t$ in (2.2) and the fact that
64 $\bar{x}_0 = \tilde{x}$, and rearranging the terms, we have

$$\sum_{t=1}^{T_s}\theta_t\mathbb{E}[\psi(\bar{x}_t) - \psi(x)] \leq \left[\frac{\gamma_s}{\alpha_s}(1 - \alpha_s) + (T_s - 1)\frac{\gamma_s p_s}{\alpha_s}\right]\mathbb{E}[\psi(\tilde{x}) - \psi(x)]$$
$$+ \mathbb{E}[V(x_0, x) - V(x_T, x)].$$

65 Now using the facts that $x^s = x_T$, $x_0 = x^{s-1}$, $\tilde{x}^s = \sum_{t=1}^{T_s}(\theta_t\bar{x}_t)/\sum_{t=1}^{T_s}\theta_t$, $\tilde{x} = \tilde{x}^{s-1}$, and the
66 convexity of $\psi$, we have

$$\sum_{t=1}^{T_s}\theta_t\mathbb{E}[\psi(\tilde{x}^s) - \psi(x)] \leq \left[\frac{\gamma_s}{\alpha_s}(1 - \alpha_s) + (T_s - 1)\frac{\gamma_s p_s}{\alpha_s}\right]\mathbb{E}[\psi(\tilde{x}^{s-1}) - \psi(x)]$$
$$+ \mathbb{E}[V(x^{s-1}, x) - V(x^s, x)],$$

67 which, in view of the fact that $\sum_{t=1}^{T_s}\theta_t = \frac{\gamma_s}{\alpha_s} + (T_s - 1)\frac{\gamma_s(\alpha_s + p_s)}{\alpha_s}$, then implies that

$$\mathcal{L}_s\mathbb{E}[\psi(\tilde{x}^s) - \psi(x)] \leq \mathcal{R}_s\mathbb{E}[\psi(\tilde{x}^{s-1}) - \psi(x)] + \mathbb{E}[V(x^{s-1}, x) - V(x^s, x)]. \qquad (A.19)$$

68 Summing over the above relations, using the convexity of $\psi$ and rearranging the terms, we then obtain
69 (A.17). □

70 With the help of Lemma 7, we are now ready to prove Theorem 1, which shows that for solving smooth
71 convex problems the Varag algorithm can achieve a fast linear rate of convergence $\mathcal{O}\{m \log \frac{D_0}{\epsilon}\}$ if
72 $m \geq D_0/\epsilon$ and an optimal sublinear rate of convergence otherwise.

73 **Proof of Theorem 1.** Let the probabilities $q_i = L_i/\sum_{i=1}^m L_i$ for $i = 1, \ldots, m$, and $\theta_t$, $\gamma_s$, $p_s$, $T_s$
74 and $\alpha_s$ be defined as in (2.2), (2.3) and (2.4). By the definition of $L_Q$ in (A.2) and the selection of $q_i$,
75 we have $L_Q = L$. Observe that both conditions in (A.11) and (A.12) are satisfied since

$$1 + \mu\gamma_s - L\alpha_s\gamma_s = 1 - L\alpha_s\gamma_s = \frac{2}{3}$$

76 and

$$p_s - \frac{L_Q\alpha_s\gamma_s}{1 + \mu\gamma_s - L\alpha_s\gamma_s} = p_s - \frac{1}{2} = 0.$$

77 Now letting $\mathcal{L}_s$ and $\mathcal{R}_s$ be defined in (A.15), we will show that $\mathcal{L}_s \geq \mathcal{R}_{s+1}$ for any $s \geq 1$. Indeed, if
78 $1 \leq s < s_0$, we have $\alpha_{s+1} = \alpha_s$, $\gamma_{s+1} = \gamma_s$, $T_{s+1} = 2T_s$, and hence

$$w_s = \mathcal{L}_s - \mathcal{R}_{s+1} = \frac{\gamma_s}{\alpha_s}\left[1 + (T_s - 1)(\alpha_s + p_s) - (1 - \alpha_s) - (2T_s - 1)p_s\right]$$
$$= \frac{\gamma_s}{\alpha_s}\left[T_s(\alpha_s - p_s)\right] = 0.$$

79 Moreover, if $s \geq s_0$, we have

$$w_s = \mathcal{L}_s - \mathcal{R}_{s+1} = \frac{\gamma_s}{\alpha_s} - \frac{\gamma_{s+1}}{\alpha_{s+1}}(1 - \alpha_{s+1}) + (T_{s_0} - 1)\left[\frac{\gamma_s(\alpha_s + p_s)}{\alpha_s} - \frac{\gamma_{s+1}p_{s+1}}{\alpha_{s+1}}\right]$$
$$= \frac{1}{12L} + \frac{(T_{s_0} - 1)[2(s - s_0 + 4) - 1]}{24L} \geq 0.$$

80 Using these observations in (A.17) iteratively, we then conclude that

$$\mathcal{L}_s\mathbb{E}[\psi(\tilde{x}^s) - \psi(x)] \leq \mathcal{R}_1\mathbb{E}[\psi(\tilde{x}^0) - \psi(x)] + \mathbb{E}[V(x^0, x) - V(x^s, x)]$$
$$\leq \frac{2}{3L}[\psi(x^0) - \psi(x)] + V(x^0, x)$$

for any $s \geq 1$, where the last identity follows from the fact that $\mathcal{R}_1 = \frac{2}{3L}$. Recalling that $D_0 := 2[\psi(x^0) - \psi(x)] + 3LV(x^0, x)$ in (2.6), now we distinguish the following two cases.

**Case 1:** if $s \leq s_0$, $\mathcal{L}_s = \frac{2^{s+1}}{3L}$. Therefore, we have

$$\mathbb{E}[\psi(\tilde{x}^s) - \psi(x)] \leq 2^{-(s+1)} D_0, \quad 1 \leq s \leq s_0.$$

**Case 2:** if $s \geq s_0$, we have

$$
\begin{aligned}
\mathcal{L}_s &= \tfrac{1}{3L\alpha_s^2} \left[ 1 + (T_s - 1)(\alpha_s + \tfrac{1}{2}) \right] \\
&= \tfrac{(s-s_0+4)(T_{s_0}-1)}{6L} + \tfrac{(s-s_0+4)^2(T_{s_0}+1)}{24L} \\
&\geq \tfrac{(s-s_0+4)^2 m}{48L},
\end{aligned}
\tag{A.20}
$$

where the last inequality follows from $T_{s_0} = 2^{\lfloor \log_2 m \rfloor + 1 - 1} \geq m/2$. Hence, we obtain

$$\mathbb{E}[\psi(\tilde{x}^s) - \psi(x)] \leq \tfrac{16 D_0}{(s-s_0+4)^2 m}, \quad s > s_0.$$

In conclusion, we have for any $x \in X$,

$$
\mathbb{E}[\psi(\tilde{x}^s) - \psi(x)] \leq
\begin{cases}
2^{-(s+1)} D_0, & 1 \leq s \leq s_0, \\
\tfrac{16 D_0}{(s-s_0+4)^2 m}, & s > s_0.
\end{cases}
\tag{A.21}
$$

In order to derive the complexity bounds in Theorem 1, let us first consider the region of relatively low accuracy and/or large number of components, i.e., $m \geq D_0/\epsilon$. In this case Varag needs to run at most $s_0$ epochs because by the first case of (A.21) we can easily check that

$$\tfrac{D_0}{2^{s_0+1}} \leq \epsilon.$$

More precisely, the number of epochs can be bounded by $S_l := \min \left\{ \log \tfrac{D_0}{\epsilon}, s_0 \right\}$. Hence the total number of gradient evaluations can be bounded by

$$
mS_l + \sum_{s=1}^{S_l} T_s = mS_l + \sum_{s=1}^{S_l} 2^{s-1} = \mathcal{O} \left\{ \min \left( m \log \tfrac{D_0}{\epsilon}, m \log m \right) \right\} = \mathcal{O} \left\{ m \log \tfrac{D_0}{\epsilon} \right\},
\tag{A.22}
$$

where the last identity follows from the assumption that $m \geq D_0/\epsilon$. Now let us consider the region for high accuracy and/or smaller number of components, i.e., $m < D_0/\epsilon$. In this case, we may need to run the algorithm for more than $s_0$ epochs. More precisely, the total number of epochs can be bounded by $S_h := \left\lceil \sqrt{\tfrac{16 D_0}{m\epsilon}} + s_0 - 4 \right\rceil$. Note that the total number of gradient evaluations needed for the first $s_0$ epochs can be bounded by $ms_0 + \sum_{s=1}^{s_0} T_s$ while the total number of gradient evaluations for the remaining epochs can be bounded by $(T_{s_0} + m)(S_h - s_0)$. As a consequence, the total number of gradient evaluations of $f_i$ can be bounded by

$$
ms_0 + \sum_{s=1}^{s_0} T_s + (T_{s_0} + m)(S_h - s_0) \leq \sum_{s=1}^{s_0} T_s + (T_{s_0} + m)S_h = \mathcal{O} \left\{ \sqrt{\tfrac{m D_0}{\epsilon}} + m \log m \right\}.
\tag{A.23}
$$

Therefore, the results of Theorem 1 follows immediately by combining these two cases. $\qquad \square$

## A.2  Convex finite-sum problems with or without strong convexity

In this subsection, we provide a unified analysis of Varag when $f$ is possibly strongly convex, i.e., $\mu \geq 0$ in (2.1). In particular, it achieves a stronger rate of convergence than other RIG methods if the condition number $L/\mu$ is very large. Below we consider four different cases and establish the convergence properties of Varag in each case.

**Lemma 8** *If $s \leq s_0$, then for any $x \in X$,*

$$\mathbb{E}[\psi(\tilde{x}^s) - \psi(x)] \leq 2^{-(s+1)} D_0, \quad 1 \leq s \leq s_0,$$

*where $D_0$ is defined in (2.6).*

107   *Proof:* In this case, we have $\alpha_s = p_s = \frac{1}{2}$, $\gamma_s = \frac{2}{3L}$, and $T_s = 2^{s-1}$. It then follows from (A.13) that

$$\frac{\gamma_s}{\alpha_s}\mathbb{E}[\psi(\bar{x}_t) - \psi(x)] + (1 + \mu\gamma_s)\mathbb{E}[V(x_t, x)] \leq \frac{\gamma_s}{2\alpha_s}\mathbb{E}[\psi(\tilde{x}) - \psi(x)] + \mathbb{E}[V(x_{t-1}, x)].$$

108   Summing up the above relation from $t = 1$ to $T_s$, we have

$$\frac{\gamma_s}{\alpha_s}\sum_{t=1}^{T_s}\mathbb{E}[\psi(\bar{x}_t) - \psi(x)] + \mathbb{E}[V(x_{T_s}, x)] + \mu\gamma_s\sum_{t=1}^{T_s}\mathbb{E}[V(x_t, x)]$$
$$\leq \frac{\gamma_s T_s}{2\alpha_s}\mathbb{E}[\psi(\tilde{x}) - \psi(x)] + \mathbb{E}[V(x_0, x)].$$

109   Note that in this case $\theta_t$ are chosen as in (2.2), i.e., $\theta_t = \frac{\gamma_s}{\alpha_s}$, $t = 1, \ldots, T_s$ in the definition of $\tilde{x}^s$, we
110   then have

$$\frac{4T_s}{3L}\mathbb{E}[\psi(\tilde{x}^s) - \psi(x)] + \mathbb{E}[V(x^s, x)] \leq \frac{4T_s}{6L}\mathbb{E}[\psi(\tilde{x}^{s-1}) - \psi(x)] + \mathbb{E}[V(x^{s-1}, x)]$$
$$= \frac{4T_{s-1}}{3L}\mathbb{E}[\psi(\tilde{x}^{s-1}) - \psi(x)] + \mathbb{E}[V(x^{s-1}, x)],$$

111   where we use the facts that $\tilde{x} = \tilde{x}^{s-1}$, $x_0 = x^{s-1}$, and $x^s = x_{T_s}$ in the epoch $s$ and the parameter
112   settings in (2.3). Applying this inequality recursively, we then have

$$\frac{4T_s}{3L}\mathbb{E}[\psi(\tilde{x}^s) - \psi(x)] + \mathbb{E}[V(x^s, x)] \leq \frac{2}{3L}\mathbb{E}[\psi(\tilde{x}^0) - \psi(x)] + V(x^0, x)$$
$$= \frac{2}{3L}\mathbb{E}[\psi(x^0) - \psi(x)] + V(x^0, x). \qquad \text{(A.24)}$$

113   By plugging $T_s = 2^{s-1}$ into the above inequality, we obtain the result.     $\square$

114   **Lemma 9** *If $s \geq s_0$ and $m \geq \frac{3L}{4\mu}$,*

$$\mathbb{E}[\psi(\tilde{x}^s) - \psi(x^*)] \leq \left(\tfrac{4}{5}\right)^s D_0,$$

115   *where $x^*$ is an optimal solution of* (1.1).

116   *Proof:* In this case, we have $\alpha_s = p_s = \frac{1}{2}$, $\gamma_s = \gamma = \frac{2}{3L}$, and $T_s \equiv T_{s_0} = 2^{s_0-1}, s \geq s_0$. It then
117   follows from (A.13) that

$$\frac{4}{3L}\mathbb{E}[\psi(\bar{x}_t) - \psi(x)] + (1 + \frac{2\mu}{3L})\mathbb{E}[V(x_t, x)] \leq \frac{2}{3L}\mathbb{E}[\psi(\tilde{x}) - \psi(x)] + \mathbb{E}[V(x_{t-1}, x)].$$

118   Multiplying both sides of the above inequality by $\Gamma_{t-1} = (1 + \frac{2\mu}{3L})^{t-1}$, we obtain

$$\frac{4}{3L}\Gamma_{t-1}\mathbb{E}[\psi(\bar{x}_t) - \psi(x)] + \Gamma_t\mathbb{E}[V(x_t, x)] \leq \frac{2}{3L}\Gamma_{t-1}\mathbb{E}[\psi(\tilde{x}) - \psi(x)] + \Gamma_{t-1}\mathbb{E}[V(x_{t-1}, x)].$$

119   Note that $\theta_t$ are chosen as in (2.7) when $s \geq s_0$, i.e., $\theta_t = \Gamma_{t-1} = (1 + \frac{2\mu}{3L})^{t-1}$, $t = 1, \ldots, T_s$,
120   $s \geq s_0$. Summing up the above inequality for $t = 1, \ldots, T_s$ we have

$$\frac{4}{3L}\sum_{t=1}^{T_s}\theta_t\mathbb{E}[\psi(\bar{x}_t) - \psi(x)] + \Gamma_{T_s}\mathbb{E}[V(x_{T_s}, x)]$$
$$\leq \frac{2}{3L}\sum_{t=1}^{T_s}\theta_t\mathbb{E}[\psi(\tilde{x}) - \psi(x)] + \mathbb{E}[V(x_0, x)], \; s \geq s_0.$$

121   Observe that for $s \geq s_0$, $m \geq T_s \equiv T_{s_0} = 2^{\lfloor \log_2 m \rfloor} \geq m/2$, and hence that

$$\Gamma_{T_s} = (1 + \tfrac{2\mu}{3L})^{T_s} = (1 + \tfrac{2\mu}{3L})^{T_{s_0}} \geq 1 + \tfrac{2\mu T_{s_0}}{3L} \geq 1 + \tfrac{T_{s_0}}{2m} \geq \tfrac{5}{4}, \; \forall s \geq s_0, \qquad \text{(A.25)}$$

122   and using the facts that $\tilde{x}^s = \sum_{t=1}^{T_s}(\theta_t\bar{x}_t)/\sum_{t=1}^{T_s}\theta_t$, $\tilde{x} = \tilde{x}^{s-1}$, $x_0 = x^{s-1}$, and $x_{T_s} = x^s$ in the $s$
123   epoch, and $\psi(\tilde{x}^s) - \psi(x^*) \geq 0$, we conclude from the above inequalities that

$$\frac{5}{4}\left\{\frac{2}{3L}\mathbb{E}[\psi(\tilde{x}^s) - \psi(x^*)] + (\sum_{t=1}^{T_s}\theta_t)^{-1}\mathbb{E}[V(x^s, x^*)]\right\}$$
$$\leq \frac{2}{3L}\mathbb{E}[\psi(\tilde{x}^{s-1}) - \psi(x^*)] + (\sum_{t=1}^{T_s}\theta_t)^{-1}\mathbb{E}[V(x^{s-1}, x^*)], s \geq s_0.$$

124   Applying this relation recursively for $s \geq s_0$, we then obtain

$$\frac{2}{3L}\mathbb{E}[\psi(\tilde{x}^s) - \psi(x^*)] + (\sum_{t=1}^{T_s}\theta_t)^{-1}\mathbb{E}[V(x^s, x^*)]$$
$$\leq \left(\tfrac{4}{5}\right)^{s-s_0}\left\{\frac{2}{3L}\mathbb{E}[\psi(\tilde{x}^{s_0}) - \psi(x^*)] + (\sum_{t=1}^{T_s}\theta_t)^{-1}\mathbb{E}[V(x^{s_0}, x^*)]\right\}$$
$$\leq \left(\tfrac{4}{5}\right)^{s-s_0}\left\{\frac{2}{3L}\mathbb{E}[\psi(\tilde{x}^{s_0}) - \psi(x^*)] + \frac{1}{T_{s_0}}\mathbb{E}[V(x^{s_0}, x^*)]\right\},$$

125   where the last inequality follows from $\sum_{t=1}^{T_s}\theta_t \geq T_s = T_{s_0}$. Plugging (A.24) into the above
126   inequality, we have

$$\mathbb{E}[\psi(\tilde{x}^s) - \psi(x^*)] \leq \left(\tfrac{4}{5}\right)^{s-s_0}\frac{D_0}{2T_{s_0}} = \left(\tfrac{4}{5}\right)^{s-s_0}\frac{D_0}{2^{s_0}} \leq \left(\tfrac{4}{5}\right)^s D_0, \; s \geq s_0.$$

127     $\square$

128 **Lemma 10** *If* $s_0 < s \le s_0 + \sqrt{\frac{12L}{m\mu}} - 4$ *and* $m < \frac{3L}{4\mu}$, *then for any* $x \in X$,

$$\mathbb{E}[\psi(\tilde{x}^s) - \psi(x)] \le \frac{16 D_0}{(s-s_0+4)^2 m}.$$

129 *Proof:* In this case, $\frac{1}{2} \ge \frac{2}{s-s_0+4} \ge \sqrt{\frac{m\mu}{3L}}$. Therefore, we set $\theta_t$ as in (2.2), $\alpha_s = \frac{2}{s-s_0+4}, p_s = \frac{1}{2}$,
130 $\gamma_s = \frac{1}{3L\alpha_s}$, and $T_s \equiv T_{s_0}$. Observe that the parameter setting in this case is the same as the smooth
131 case in Theorem 1. Hence, by following the same procedure as in the proof of Theorem 1, we can
132 obtain

$$
\begin{aligned}
\mathcal{L}_s \mathbb{E}[\psi(\tilde{x}^s) - \psi(x)] + \mathbb{E}[V(x^s, x)] &\le \mathcal{R}_{s_0+1}\mathbb{E}[\psi(\tilde{x}^{s_0}) - \psi(x)] + \mathbb{E}[V(x^{s_0}, x)] \\
&\le \mathcal{L}_{s_0}\mathbb{E}[\psi(\tilde{x}^{s_0}) - \psi(x)] + \mathbb{E}[V(x^{s_0}, x)] \\
&\le \tfrac{D_0}{3L},
\end{aligned}
\tag{A.26}
$$

133 where the last inequality follows from the fact that $\mathcal{L}_{s_0} \ge \frac{2T_{s_0}}{3L}$ and the relation in (A.24). The result
134 then follows by noting that $\mathcal{L}_s \ge \frac{(s-s_0+4)^2 m}{48L}$ (see (A.20)). $\qquad\square$

135 **Lemma 11** *If* $s > \bar{s}_0 := s_0 + \sqrt{\frac{12L}{m\mu}} - 4$ *and* $m < \frac{3L}{4\mu}$, *then*

$$\mathbb{E}[\psi(\tilde{x}^s) - \psi(x^*)] \le \left(1 + \sqrt{\frac{\mu}{3mL}}\right)^{\frac{-m(s-\bar{s}_0)}{2}} \frac{D_0}{3L/4\mu}, \tag{A.27}$$

136 *where* $x^*$ *is an optimal solution of* (1.1).

137 *Proof:* In this case, $\frac{1}{2} \ge \sqrt{\frac{m\mu}{3L}} \ge \frac{2}{s-s_0+4}$. Therefore, we use constant step-size policy that
138 $\alpha_s \equiv \sqrt{\frac{m\mu}{3L}}, p_s \equiv \frac{1}{2}, \gamma_s \equiv \frac{1}{3L\alpha_s} = \frac{1}{\sqrt{3mL\mu}}$, and $T_s \equiv T_{s_0}$. Also note that in this case $\theta_t$ are chosen
139 as in (2.7). Multiplying both sides of (A.13) by $\Gamma_{t-1} = (1+\mu\gamma_s)^{t-1}$, we obtain

$$
\begin{aligned}
\tfrac{\gamma_s}{\alpha_s}\Gamma_{t-1}\mathbb{E}[\psi(\bar{x}_t) - \psi(x)] + \Gamma_t \mathbb{E}[V(x_t, x)] &\le \tfrac{\Gamma_{t-1}\gamma_s}{\alpha_s}(1-\alpha_s-p_s)\mathbb{E}[\psi(\bar{x}_{t-1}) - \psi(x)] \\
&\quad + \tfrac{\Gamma_{t-1}\gamma_s p_s}{\alpha_s}\mathbb{E}[\psi(\tilde{x}) - \psi(x)] + \Gamma_{t-1}\mathbb{E}[V(x_{t-1}, x)].
\end{aligned}
$$

140 Summing up the above inequality from $t = 1, \ldots, T_s$ and using the fact that $\bar{x}_0 = \tilde{x}$, we arrive at

$$
\begin{aligned}
&\tfrac{\gamma_s}{\alpha_s}\sum_{t=1}^{T_s}\theta_t\mathbb{E}[\psi(\bar{x}_t) - \psi(x)] + \Gamma_{T_s}\mathbb{E}[V(x_{T_s}, x)] \\
&\quad \le \tfrac{\gamma_s}{\alpha_s}\left[1 - \alpha_s - p_s + p_s\sum_{t=1}^{T_s}\Gamma_{t-1}\right]\mathbb{E}[\psi(\tilde{x}) - \psi(x)] + \mathbb{E}[V(x_0, x)].
\end{aligned}
$$

141 Now using the facts that $x^s = x_{T_s}$, $x_0 = x^{s-1}$, $\tilde{x}^s = \sum_{t=1}^{T_s}(\theta_t \bar{x}_t)/\sum_{t=1}^{T_s}\theta_t$, $\tilde{x} = \tilde{x}^{s-1}$, $T_s = T_{s_0}$
142 and the convexity of $\psi$, we obtain

$$
\begin{aligned}
&\tfrac{\gamma_s}{\alpha_s}\sum_{t=1}^{T_{s_0}}\theta_t\mathbb{E}[\psi(\tilde{x}^s) - \psi(x)] + \Gamma_{T_{s_0}}\mathbb{E}[V(x^s, x)] \\
&\quad \le \tfrac{\gamma_s}{\alpha_s}\left[1 - \alpha_s - p_s + p_s\sum_{t=1}^{T_{s_0}}\Gamma_{t-1}\right]\mathbb{E}[\psi(\tilde{x}^{s-1}) - \psi(x)] + \mathbb{E}[V(x^{s-1}, x)]
\end{aligned}
\tag{A.28}
$$

143 for any $s > \bar{s}_0$. Moreover, we have

$$
\begin{aligned}
\sum_{t=1}^{T_{s_0}}\theta_t &= \Gamma_{T_{s_0}-1} + \sum_{t=1}^{T_{s_0}-1}(\Gamma_{t-1} - (1-\alpha_s-p_s)\Gamma_t) \\
&= \Gamma_{T_{s_0}}(1-\alpha_s-p_s) + \sum_{t=1}^{T_{s_0}}(\Gamma_{t-1} - (1-\alpha_s-p_s)\Gamma_t) \\
&= \Gamma_{T_{s_0}}(1-\alpha_s-p_s) + [1 - (1-\alpha_s-p_s)(1+\mu\gamma_s)]\sum_{t=1}^{T_{s_0}}\Gamma_{t-1}.
\end{aligned}
$$

144 Observe that for any $T > 1$ and $0 \le \delta T \le 1$, $(1+\delta)^T \le 1 + 2T\delta$, $\alpha_s = \sqrt{\frac{m\mu}{3L}} \ge \sqrt{\frac{T_{s_0}\mu}{3L}}$ and
145 hence that

$$
\begin{aligned}
1 - (1-\alpha_s-p_s)(1+\mu\gamma_s) &\ge (1+\mu\gamma_s)(\alpha_s - \mu\gamma_s + p_s) \\
&\ge (1+\mu\gamma_s)(T_{s_0}\mu\gamma_s - \mu\gamma_s + p_s) \\
&= p_s(1+\mu\gamma_s)[2(T_{s_0}-1)\mu\gamma_s + 1] \\
&\ge p_s(1+\mu\gamma_s)^{T_{s_0}} = p_s\Gamma_{T_{s_0}}.
\end{aligned}
$$

146 Then we conclude that $\sum_{t=1}^{T_{s_0}}\theta_t \geq \Gamma_{T_{s_0}}\left[1-\alpha_s-p_s+p_s\sum_{t=1}^{T_{s_0}}\Gamma_{t-1}\right]$. Together with (A.28) and

147 the fact that $\psi(\tilde{x}^s)-\psi(x^*) \geq 0$, we have

$$\Gamma_{T_{s_0}}\left\{\tfrac{\gamma_s}{\alpha_s}\left[1-\alpha_s-p_s+p_s\sum_{t=1}^{T_{s_0}}\Gamma_{t-1}\right]\mathbb{E}[\psi(\tilde{x}^s)-\psi(x^*)]+\mathbb{E}[V(x^s,x^*)]\right\}$$
$$\leq \tfrac{\gamma_s}{\alpha_s}\left[1-\alpha_s-p_s+p_s\sum_{t=1}^{T_{s_0}}\Gamma_{t-1}\right]\mathbb{E}[\psi(\tilde{x}^{s-1})-\psi(x^*)]+\mathbb{E}[V(x^{s-1},x^*)].$$

148 Applying the above relation recursively for $s > \bar{s}_0 = s_0+\sqrt{\tfrac{12L}{m\mu}}-4$, and also noting that

149 $\Gamma_t = (1+\mu\gamma_s)^t$ and the constant step-size policy in this case, we obtain

$$\tfrac{\gamma_s}{\alpha_s}\left[1-\alpha_s-p_s+p_s\sum_{t=1}^{T_{s_0}}\Gamma_{t-1}\right]\mathbb{E}[\psi(\tilde{x}^s)-\psi(x^*)]+\mathbb{E}[V(x^s,x^*)]$$
$$\leq (1+\mu\gamma_s)^{-T_{s_0}(s-\bar{s}_0)}\left\{\tfrac{\gamma_s}{\alpha_s}\left[1-\alpha_s-p_s+p_s\sum_{t=1}^{T_{s_0}}\Gamma_{t-1}\right]\right.$$
$$\left.\mathbb{E}[\psi(\tilde{x}^{\bar{s}_0})-\psi(x^*)]+\mathbb{E}[V(x^{\bar{s}_0},x^*)]\right\}.$$

150 According to the parameter settings in this case, i.e., $\alpha_s \equiv \sqrt{\tfrac{m\mu}{3L}}, p_s \equiv \tfrac{1}{2}, \gamma_s \equiv \tfrac{1}{3L\alpha_s}=\tfrac{1}{\sqrt{3mL\mu}}$,

151 and $\bar{s}_0 = s_0+\sqrt{\tfrac{12L}{m\mu}}-4$, we have $\tfrac{\gamma_s}{\alpha_s}\left[1-\alpha_s-p_s+p_s\sum_{t=1}^{T_{s_0}}\Gamma_{t-1}\right] \geq \tfrac{\gamma_s p_s T_{s_0}}{\alpha_s}=\tfrac{T_{s_0}}{2m\mu}=$

152 $\tfrac{(\bar{s}_0-s_0+4)^2 T_{s_0}}{24L}$. Using this observation in the above inequality, we then conclude that

$$\mathbb{E}[\psi(\tilde{x}^s)-\psi(x^*)] \leq (1+\mu\gamma_s)^{-T_{s_0}(s-\bar{s}_0)}\left[\mathbb{E}[\psi(\tilde{x}^{\bar{s}_0})-\psi(x^*)]+\tfrac{24L}{(\bar{s}_0-s_0+4)^2 T_{s_0}}\mathbb{E}[V(x^{\bar{s}_0},x^*)]\right]$$
$$\leq (1+\mu\gamma_s)^{-T_{s_0}(s-\bar{s}_0)}\tfrac{24L}{(\bar{s}_0-s_0+4)^2 T_{s_0}}\left[\mathcal{L}_{\bar{s}_0}\mathbb{E}[\psi(\tilde{x}^{\bar{s}_0})-\psi(x^*)]+\mathbb{E}[V(x^{\bar{s}_0},x^*)]\right]$$
$$\leq (1+\mu\gamma_s)^{-T_{s_0}(s-\bar{s}_0)}\tfrac{24L}{(\bar{s}_0-s_0+4)^2 T_{s_0}}\tfrac{D_0}{3L}$$
$$\leq (1+\mu\gamma_s)^{-T_{s_0}(s-\bar{s}_0)}\tfrac{16D_0}{(\bar{s}_0-s_0+4)^2 m}$$
$$= (1+\mu\gamma_s)^{-T_{s_0}(s-\bar{s}_0)}\tfrac{D_0}{3L/4\mu},$$

153 where the second inequality follows from the fact that $\mathcal{L}_{\bar{s}_0} \geq \tfrac{(\bar{s}_0-s_0+4)^2 T_{s_0}}{24L}=\tfrac{T_{s_0}}{2m\mu}$ due to (A.20),

154 the third inequality follows from (A.26) in Case 3, and last inequality follows from $T_{s_0}=2^{\lfloor \log_2 m\rfloor} \geq$

155 $m/2$. $\qquad\qquad\square$

156 Putting the above four technical results together, we are ready to prove Theorem 2 for Varag solving

157 (1.1) when (1.1) is possibly strongly convex.

158 **Proof of Theorem 2.** Suppose that the probabilities $q_i$'s are set to $L_i/\sum_{i=1}^{m}L_i$ for $i=1,\ldots,m$.

159 Moreover, let us denote $s_0 := \lfloor\log m\rfloor+1$ and assume that the weights $\{\theta_t\}$ are set to (2.2) if

160 $1 \leq s \leq s_0$ or $s_0 < s \leq s_0+\sqrt{\tfrac{12L}{m\mu}}-4$, $m < \tfrac{3L}{4\mu}$. Otherwise, they are set to (2.7). If the parameters

161 $\{T_s\}$, $\{\gamma_s\}$ and $\{p_s\}$ set to (2.3) with $\{\alpha_s\}$ given by (2.8), then we have

$$\mathbb{E}[\psi(\tilde{x}^s)-\psi(x^*)] \leq \begin{cases} 2^{-(s+1)}D_0, & 1 \leq s \leq s_0, \\ \left(\tfrac{4}{5}\right)^s D_0, & s > s_0, \text{ and } m \geq \tfrac{3L}{4\mu}, \\ \tfrac{16D_0}{(s-s_0+4)^2 m}, & s_0 < s \leq s_0+\sqrt{\tfrac{12L}{m\mu}}-4 \text{ and } m < \tfrac{3L}{4\mu}, \\ \left(1+\sqrt{\tfrac{\mu}{3mL}}\right)^{\tfrac{-m(s-\bar{s}_0)}{2}}\tfrac{D_0}{3L/4\mu}, & s_0+\sqrt{\tfrac{12L}{m\mu}}-4 = \bar{s}_0 < s \text{ and } m < \tfrac{3L}{4\mu}, \end{cases}$$
$$\text{(A.29)}$$

162 where $x^*$ is an optimal solution of (1.1) and $D_0$ is defined as in (2.6).

163 Now we are ready to provide the proof for the complexity results presented in Theorem 2. Firstly, it

164 is clear that the first case and the third case corresponds to the results of the smooth case discussed

165 in Theorem 1. As a consequence, the total number of gradient evaluations can also be bounded by

166 (A.22) and (A.23), respectively. Secondly, for the second case of (A.29), it is easy to check that

167 Varag needs to run at most $S := \mathcal{O}\{\log D_0/\epsilon\}$ epochs, and hence the total number of gradient
168 evaluations can be bounded by

$$mS + \sum_{s=1}^{S} T_s \le 2mS = \mathcal{O}\left\{m \log \frac{D_0}{\epsilon}\right\}. \tag{A.30}$$

169 Finally, let us consider the last case of (A.29). Since Varag only needs to run at most $S' =$
170 $\bar{s}_0 + 2\sqrt{\frac{3L}{m\mu}} \log \frac{D_0/\epsilon}{3L/4\mu}$ epochs in this case, the total number of gradient evaluations can be bounded
171 by

$$\sum_{s=1}^{S'} (m + T_s) = \sum_{s=1}^{s_0} (m + T_s) + \sum_{s=s_0+1}^{\bar{s}_0} (m + T_{s_0}) + (m + T_{s_0})(S' - \bar{s}_0)$$

$$\le 2m \log m + 2m(\sqrt{\frac{12L}{m\mu}} - 4) + 4m\sqrt{\frac{3L}{m\mu}} \log \frac{D_0/\epsilon}{3L/4\mu}$$

$$= \mathcal{O}\left\{m \log m + \sqrt{\frac{mL}{\mu}} \log \frac{D_0/\epsilon}{3L/4\mu}\right\}, \tag{A.31}$$

172 Therefore, the results of Theorem 2 follows immediately from the above discussion. $\qquad\square$

### A.3 Convex finite-sum optimization under error bound

174 In this section, we consider a class of convex finite-sum optimization problems that satisfies the error
175 bound condition described in (2.10), and establish the convergence results for applying Varag to
176 solve it.

177 **Proof of Theorem 3.** Similar to the smooth case, according to (A.17), for any $x \in X$, we have

$$\mathcal{L}_s \mathbb{E}[\psi(\tilde{x}^s) - \psi(x)] \le \mathcal{R}_1 \mathbb{E}[\psi(\tilde{x}^0) - \psi(x)] + \mathbb{E}[V(x^0, x) - V(x^s, x)]$$

$$\le \mathcal{R}_1[\psi(x^0) - \psi(x)] + V(x^0, x).$$

178 Then we use $x^*$ to replace $x$ and use the relation of (2.10) to obtain

$$\mathcal{L}_s \mathbb{E}[\psi(\tilde{x}^s) - \psi(x^*)] \le \mathcal{R}_1[\psi(x^0) - \psi(x^*)] + \frac{1}{u}[\psi(x) - \psi(x^*)].$$

179 Now, we compute $\mathcal{L}_s$ and $\mathcal{R}_1$. According to (A.20), we have $\mathcal{L}_s \ge \frac{(s-s_0+4)^2(T_{s_0}+1)}{24L}$. We have
180 $\mathcal{R}_1 = \frac{2T_1}{3L}$ by plugging the parameters $\gamma_1, p_1, \alpha_1$ and $T_1$ into (A.15).

181 Thus, we prove (2.12) as follows (recall that $s_0 = 4$ and $s = s_0 + 4\sqrt{\frac{L}{\bar{\mu}m}}$):

$$\mathbb{E}[\psi(\tilde{x}^s) - \psi(x^*)] \le \frac{16T_1 + 24L/\bar{\mu}}{(s-s_0+4)^2 T_1 2^{s_0-1}}[\psi(x^0) - \psi(x^*)]$$

$$\le \frac{16 + 24L/(\bar{\mu}T_1)}{(s-s_0+4)^2 2^{s_0-1}}[\psi(x^0) - \psi(x^*)]$$

$$\le \frac{5}{16} \frac{L/(\bar{\mu}T_1)}{1+L/(\bar{\mu}m)}[\psi(x^0) - \psi(x^*)]$$

$$\le \frac{5}{16}[\psi(x^0) - \psi(x^*)],$$

182 where the last inequality follows from $T_1 = \min\{m, \frac{L}{\bar{\mu}}\}$.

183 Finally, we plug $k = \log \frac{\psi(x^0) - \psi(x^*)}{\epsilon}, s_0 = 4, s = s_0 + 4\sqrt{\frac{L}{\bar{\mu}m}}$ and $T_1 = \min\{m, \frac{L}{\bar{\mu}}\}$ to prove
184 (2.13):

$$\bar{N} := k(\sum_s (m + T_s)) \le k(ms + T_1 2^{s_0}(s - s_0 + 1)) = \mathcal{O}\left(m + \sqrt{\frac{mL}{\bar{\mu}}}\right) \log \frac{\psi(x^0) - \psi(x^*)}{\epsilon}.$$

185 $\qquad\square$

## B Varag for stochastic finite-sum optimization

187 In this section, we consider the stochastic finite-sum optimization and online learning problems,
188 where only noisy gradient information of $f_i$ can be accessed via the SFO oracle, and provide the
189 proof of Theorem 4.

190 Before proving Theorem 4, we need to establish some key technical results in the following lemmas.
191 First, we rewrite Lemma 3 under the stochastic setting. Lemma 12 below shows that $G_t$ updated
192 according to Algorithm 2 is an unbiased estimator of $\nabla f(\underline{x}_t)$ and its variance is upper bounded.

193 **Lemma 12** *Conditionally on $x_1, \ldots, x_t$,*

$$\mathbb{E}[\delta_t] = 0, \tag{B.1}$$

$$\mathbb{E}[\|\delta_t\|_*^2] \le 2L_Q[f(\tilde{x}) - f(\underline{x}_t) - \langle \nabla f(\underline{x}_t), \tilde{x} - \underline{x}_t \rangle] + \sum_{i=1}^m \frac{\sigma^2}{q_i m^2 b_s} + \sum_{i=1}^m \frac{2\sigma^2}{q_i m^2 B_s} + \frac{2\sigma^2}{m B_s}, \tag{B.2}$$

194 *where $\delta_t = G_t - \nabla f(\underline{x}_t)$ and $G_t = \frac{1}{q_{i_t} m b_s} \sum_{k=1}^{b_s} \big( G_{i_t}(\underline{x}_t, \xi_k^s) - G_{i_t}(\tilde{x}) \big) + \tilde{g}$ (see Line 2.17 of*
195 *Algorithm 2).*

196 *Proof:* Take the expectation with respect to $i_t$ and $[\xi] := \{\xi_k\}_{k=1}^{b_s}$ conditionally on $x_1, \ldots, x_t$, we
197 obtain

$$\mathbb{E}_{i_t, [\xi]} \Big[ \frac{1}{m q_{i_t} b_s} \sum_{k=1}^{b_s} G_{i_t}(\underline{x}_t, \xi_k) - \frac{1}{m q_{i_t}} G_{i_t}(\tilde{x}) + \frac{1}{m} \sum_{i=1}^m G_i(\tilde{x}) - \nabla f(\underline{x}_t) \Big]$$

$$= \mathbb{E}_{i_t} \Big[ \frac{1}{m q_{i_t}} \nabla f_{i_t}(\underline{x}_t) - \frac{1}{m q_{i_t}} G_{i_t}(\tilde{x}) + \frac{1}{m} \sum_{i=1}^m G_i(\tilde{x}) - \nabla f(\underline{x}_t) \Big]$$

$$= 0,$$

198 where the first equality follows from (2.14).

199 Moreover, we have

$$\mathbb{E}[\|\delta_t\|_*^2] = \mathbb{E}\Big[ \Big\| \frac{1}{m q_{i_t} b_s} \sum_{k=1}^{b_s} G_{i_t}(\underline{x}_t, \xi_k) - \frac{1}{m q_{i_t}} G_{i_t}(\tilde{x}) + \frac{1}{m} \sum_{i=1}^m G_i(\tilde{x}) - \nabla f(\underline{x}_t) \Big\|_*^2 \Big]$$

$$= \mathbb{E}\Big[ \Big\| \frac{1}{m q_{i_t}} \big( \nabla f_{i_t}(\underline{x}_t) - \nabla f_{i_t}(\tilde{x}) \big) + \nabla f(\tilde{x}) - \nabla f(\underline{x}_t) \Big\|_*^2 \Big]$$

$$+ \mathbb{E}\Big[ \Big\| \frac{1}{m q_{i_t} b_s} \sum_{k=1}^{b_s} G_{i_t}(\underline{x}_t, \xi_k) - \frac{1}{m q_{i_t}} \nabla f_{i_t}(\underline{x}_t) \Big\|_*^2 \Big]$$

$$+ \mathbb{E}\Big[ \Big\| \frac{1}{m q_{i_t}} \nabla f_{i_t}(\tilde{x}) - \frac{1}{m q_{i_t}} G_{i_t}(\tilde{x}) + \frac{1}{m} \sum_{i=1}^m G_i(\tilde{x}) - \frac{1}{m} \sum_{i=1}^m \nabla f_i(\tilde{x}) \Big\|_*^2 \Big]$$

$$\le \mathbb{E}\Big[ \frac{1}{m^2 q_{i_t}^2} \big\| \nabla f_{i_t}(\underline{x}_t) - \nabla f_{i_t}(\tilde{x}) \big\|_*^2 \Big] + \sum_{i=1}^m \frac{\sigma^2}{q_i m^2 b_s}$$

$$+ 2\mathbb{E}\Big[ \Big\| \frac{1}{m q_{i_t}} \nabla f_{i_t}(\tilde{x}) - \frac{1}{m q_{i_t}} G_{i_t}(\tilde{x}) \Big\|_*^2 \Big] + 2\mathbb{E}\Big[ \Big\| \frac{1}{m} \sum_{i=1}^m G_i(\tilde{x}) - \frac{1}{m} \sum_{i=1}^m \nabla f_i(\tilde{x}) \Big\|_*^2 \Big]$$

$$\le \mathbb{E}\Big[ \frac{1}{m^2 q_{i_t}^2} \big\| \nabla f_{i_t}(\underline{x}_t) - \nabla f_{i_t}(\tilde{x}) \big\|_*^2 \Big] + \sum_{i=1}^m \frac{\sigma^2}{q_i m^2 b_s} + \sum_{i=1}^m \frac{2\sigma^2}{q_i m^2 B_s} + \frac{2\sigma^2}{m B_s},$$

200 where the last inequality uses (2.15) and in view of relation (A.3) (with $x$ and $x^*$ replaced by $\tilde{x}$ and
201 $\underline{x}_t$), then implies (B.2). $\qquad\square$

202 We are now ready to rewrite Lemma 6 under the stochastic setting.

203 **Lemma 13** *Assume that $\alpha_s \in [0, 1]$, $p_s \in [0, 1]$ and $\gamma_s > 0$ satisfy (A.11) and (A.12). Then,*
204 *conditional on $x_1, \ldots, x_{t-1}$, we have*

$$\mathbb{E}[\psi(\bar{x}_t) + \tfrac{\alpha_s(1+\mu\gamma_s)}{\gamma_s} V(x_t, x)] \le (1 - \alpha_s - p_s)\psi(\bar{x}_{t-1}) + p_s\psi(\tilde{x}) + \alpha_s\psi(x) + \tfrac{\alpha_s}{\gamma_s} V(x_{t-1}, x)$$

$$+ \tfrac{\alpha_s \gamma_s}{2(1+\mu\gamma_s - L\alpha_s\gamma_s)} \Big( \sum_{i=1}^m \tfrac{\sigma^2}{q_i m^2 b_s} + \sum_{i=1}^m \tfrac{2\sigma^2}{q_i m^2 B_s} + \tfrac{2\sigma^2}{m B_s} \Big) \tag{B.3}$$

205 *for any $x \in X$.*

*Proof:* Similar to the proof of Lemma 6, in view of the smoothness and (strong) convexity of $f$, we recall the result in (A.14), i.e.,

$$f(\bar{x}_t) \leq (1 - \alpha_s - p_s)f(\bar{x}_{t-1}) + \alpha_s \left[\psi(x) - h(x_t) + \tfrac{1}{\gamma_s}V(x_{t-1}, x) - \tfrac{1+\mu\gamma_s}{\gamma_s}V(x_t, x)\right]$$
$$+ p_s l_f(\underline{x}_t, \tilde{x}) + \tfrac{\alpha_s\gamma_s\|\delta_t\|_*^2}{2(1+\mu\gamma_s - L\alpha_s\gamma_s)} - \alpha_s\langle\delta_t, x_{t-1}^+ - x\rangle. \tag{B.4}$$

Also note that by (B.1), (B.2), (A.12) and the convexity of $f$, we have, conditional on $x_1, \ldots, x_{t-1}$,

$$p_s l_f(\underline{x}_t, \tilde{x}) + \tfrac{\alpha_s\gamma_s\mathbb{E}[\|\delta_t\|_*^2]}{2(1+\mu\gamma_s - L\alpha_s\gamma_s)} + \alpha_s\mathbb{E}[\langle\delta_t, x_{t-1}^+ - x\rangle]$$
$$\leq p_s l_f(\underline{x}_t, \tilde{x}) + \tfrac{L_Q\alpha_s\gamma_s}{1+\mu\gamma_s - L\alpha_s\gamma_s}[f(\tilde{x}) - l_f(\underline{x}_t, \tilde{x})]$$
$$+ \tfrac{\alpha_s\gamma_s}{2(1+\mu\gamma_s - L\alpha_s\gamma_s)}\Big(\textstyle\sum_{i=1}^m \tfrac{\sigma^2}{q_i m^2 b_s} + \sum_{i=1}^m \tfrac{2\sigma^2}{q_i m^2 B_s} + \tfrac{2\sigma^2}{m B_s}\Big)$$
$$\leq \Big(p_s - \tfrac{L_Q\alpha_s\gamma_s}{1+\mu\gamma_s - L\alpha_s\gamma_s}\Big)l_f(\underline{x}_t, \tilde{x}) + \tfrac{L_Q\alpha_s\gamma_s}{1+\mu\gamma_s - L\alpha_s\gamma_s}f(\tilde{x})$$
$$+ \tfrac{\alpha_s\gamma_s}{2(1+\mu\gamma_s - L\alpha_s\gamma_s)}\Big(\textstyle\sum_{i=1}^m \tfrac{\sigma^2}{q_i m^2 b_s} + \sum_{i=1}^m \tfrac{2\sigma^2}{q_i m^2 B_s} + \tfrac{2\sigma^2}{m B_s}\Big)$$
$$\leq p_s f(\tilde{x}) + \tfrac{\alpha_s\gamma_s}{2(1+\mu\gamma_s - L\alpha_s\gamma_s)}\Big(\textstyle\sum_{i=1}^m \tfrac{\sigma^2}{q_i m^2 b_s} + \sum_{i=1}^m \tfrac{2\sigma^2}{q_i m^2 B_s} + \tfrac{2\sigma^2}{m B_s}\Big).$$

Moreover, by convexity of $h$, we have $h(\bar{x}_t) \leq (1 - \alpha_s - p_s)h(\bar{x}_{t-1}) + \alpha_s h(x_t) + p_s h(\tilde{x})$. The result then follows by summing up the previous two conclusions with (B.4). $\square$

Finally, we need to rewrite the stochastic counterpart of the decrease of function value in each epoch (Lemma 7) in the following lemma.

**Lemma 14** *Assume that for each epoch $s$, $s \geq 1$, we have $\alpha_s$, $\gamma_s$, $p_s$ and $T_s$ such that (A.11)-(A.12) hold. Also, let us set $\theta_t$ as (2.2). Moreover, let $\mathcal{L}_s$, $\mathcal{R}_s$ and $w_s$ defined as in (A.15) and (A.16) respectively. Then we have*

$$\mathcal{L}_s\mathbb{E}[\psi(\tilde{x}^s) - \psi(x)] + (\textstyle\sum_{j=1}^{s-1}w_j)\mathbb{E}[\psi(\bar{x}^s) - \psi(x)]$$
$$\leq \mathcal{R}_1\mathbb{E}[\psi(\tilde{x}^0) - \psi(x)] + \mathbb{E}[V(x^0, x) - V(x^s, x)]$$
$$+ \textstyle\sum_{j=1}^s \tfrac{\gamma_j^2 T_j}{2(1+\mu\gamma_j - L\alpha_j\gamma_j)}\Big(\sum_{i=1}^m \tfrac{\sigma^2}{q_i m^2 b_j} + \sum_{i=1}^m \tfrac{2\sigma^2}{q_i m^2 B_j} + \tfrac{2\sigma^2}{m B_j}\Big) \tag{B.5}$$

*for any $x \in X$, where $\bar{x}^s$ is defined as in (A.18).*

*Proof:* Using our assumptions on $\alpha_s$, $\gamma_s$ and $p_s$, the fact that $\mu = 0$, and subtracting $\psi(x)$ from the concluding inequality (B.3) of Lemma 13, we have

$$\tfrac{\gamma_s}{\alpha_s}\mathbb{E}[\psi(\bar{x}_t) - \psi(x)] \leq \tfrac{\gamma_s}{\alpha_s}(1 - \alpha_s - p_s)\mathbb{E}[\psi(\bar{x}_{t-1}) - \psi(x)] + \tfrac{\gamma_s p_s}{\alpha_s}\mathbb{E}[\psi(\tilde{x}) - \psi(x)]$$
$$+ \mathbb{E}[V(x_{t-1}, x) - V(x_t, x)]$$
$$+ \tfrac{\gamma_s^2}{2(1+\mu\gamma_s - L\alpha_s\gamma_s)}\Big(\textstyle\sum_{i=1}^m \tfrac{\sigma^2}{q_i m^2 b_s} + \sum_{i=1}^m \tfrac{2\sigma^2}{q_i m^2 B_s} + \tfrac{2\sigma^2}{m B_s}\Big).$$

Hence following the same procedure as we did in proving Lemma 7, we can obtain (B.5). $\square$

With the help of Lemma 14, we are now ready to prove Theorem 4, which establishes the convergence properties of Varag for solving stochastic smooth finite-sum problems given in the form of (1.1).

**Proof of Theorem 4.** Let the probabilities $q_i = L_i/\sum_{i=1}^m L_i$ for $i = 1, \ldots, m$, we then have $L_Q = L$. Clearly by setting $\alpha_s$, $\gamma_s$, and $p_s$ in (2.3) and (2.4), conditions (A.11) and (A.12) are satisfied. Moreover, similar to the deterministic case, by setting $\mathcal{L}_s$ and $\mathcal{R}_s$ as in (A.15), we can show that $\mathcal{L}_s \geq \mathcal{R}_{s+1}$ for any $s \geq 1$. Using these observations in (B.5), we then conclude that

$$\mathcal{L}_s\mathbb{E}[\psi(\tilde{x}^s) - \psi(x)] \leq \mathcal{R}_1\mathbb{E}[\psi(\tilde{x}^0) - \psi(x)] + \mathbb{E}[V(x^0, x) - V(x^s, x)]$$
$$+ \textstyle\sum_{j=1}^s \tfrac{3\gamma_j^2 T_j}{4}\Big(\sum_{i=1}^m \tfrac{\sigma^2}{q_i m^2 b_j} + \sum_{i=1}^m \tfrac{2\sigma^2}{q_i m^2 B_j} + \tfrac{2\sigma^2}{m B_j}\Big)$$
$$\leq \tfrac{2}{3L}[\psi(x^0) - \psi(x)] + V(x^0, x)$$
$$+ \textstyle\sum_{j=1}^s \tfrac{T_j}{12L^2\alpha_j^2}\Big(\tfrac{C\sigma^2}{b_j} + \tfrac{2C\sigma^2}{B_j} + \tfrac{2\sigma^2}{m B_j}\Big)$$
$$\leq \tfrac{2}{3L}[\psi(x^0) - \psi(x)] + V(x^0, x)$$
$$+ \textstyle\sum_{j=1}^s \tfrac{T_j}{12L^2\alpha_j^2}\Big(\tfrac{C\sigma^2}{b_j} + \tfrac{4C\sigma^2}{B_j}\Big)$$

for any $s \geq 1$, where the second inequality follows from the fact that $\mathcal{R}_1 = \frac{2}{3L}, \gamma_s = \frac{1}{3L\alpha_s}$, and the definition $C := \sum_{i=1}^m \frac{1}{q_i m^2}$. Note that the last two terms $\frac{C\sigma^2}{b_j}$ and $\frac{4C\sigma^2}{B_j}$ are in the same order. Also note that the sampling complexity (number of calls to the SFO oracle) is bounded by $\sum_s mB_s + \sum_s T_s b_s$ and the communication complexity (CC), if in the distributed machine learning case, is bounded by $\sum_s (m + T_s)$. So we can let $B_j \equiv b_j$, then these two complexity are bounded by their first term $m\sum_s B_s$ and $mS$ respective (note that $T_s$ is always no larger than $m$). Concretely, we let

$$B_j \equiv b_j := \begin{cases} b_1(\frac{3}{2})^{j-1}, & j \leq s_0 \\ b' & j > s_0 \end{cases}. \tag{B.6}$$

Recalling that $D_0 := 2[\psi(x^0) - \psi(x)] + 3LV(x^0, x)$ in (2.6), now we distinguish the following two cases.

**Case 1:** if $s \leq s_0 = \lfloor \log m \rfloor + 1$, $\mathcal{L}_s = \frac{T_s}{3L\alpha_s^2} = \frac{2^{s+1}}{3L}$. Therefore, we have

$$\begin{aligned}
\mathbb{E}[\psi(\tilde{x}^s) - \psi(x)] &\leq 2^{-(s+1)}D_0 + 2^{-(s+1)}\sum_{j=1}^s \frac{2^{j-1}}{L}\left(\frac{C\sigma^2}{b_j} + \frac{4C\sigma^2}{B_j}\right) \\
&\leq 2^{-(s+1)}D_0 + 2^{-(s+1)}\sum_{j=1}^s \frac{5C\sigma^2 2^{j-1}}{LB_j} \\
&\leq 2^{-(s+1)}D_0 + 2^{-(s+1)}\sum_{j=1}^s (\frac{4}{3})^{j-1}\frac{5C\sigma^2}{Lb_1} \\
&\leq 2^{-(s+1)}D_0 + (\frac{2}{3})^s \frac{15C\sigma^2}{2Lb_1} \\
&= \frac{\epsilon}{2} + \frac{\epsilon}{2}, \quad 1 \leq s \leq s_0.
\end{aligned}$$

where the last equality holds when $s = \log\frac{D_0}{\epsilon}$ and $b_1 = (\frac{2}{3})^s \frac{15C\sigma^2}{L\epsilon}$.

In this case, Varag needs to run at most $S_l := \min\left\{\log\frac{D_0}{\epsilon}, s_0\right\}$ epochs. Hence, the sampling complexity (number of calls to the SFO oracle) is bounded by

$$\sum_{s=1}^{S_l}(mB_s + T_s b_s) \leq 2m\sum_{s=1}^{S_l} b_1(\frac{3}{2})^{s-1} \leq 4mb_1(\frac{3}{2})^{S_l} = \mathcal{O}\left\{\frac{mC\sigma^2}{L\epsilon}\right\}, \tag{B.7}$$

and the communication complexity (CC), if in the distributed machine learning case, is bounded by

$$\sum_{s=1}^{S_l}(m + T_s) \leq 2mS_l = \mathcal{O}\left\{m\log\frac{D_0}{\epsilon}\right\}, \quad m \geq \frac{D_0}{\epsilon}. \tag{B.8}$$

**Case 2:** if $s \geq s_0$, $\mathcal{L}_s \geq \frac{(s-s_0+4)^2 T_{s_0}}{24L}$. Therefore, we have

$$\begin{aligned}
\mathbb{E}[\psi(\tilde{x}^s) - \psi(x)] &\leq \frac{8D_0}{(s-s_0+4)^2 T_{s_0}} + \frac{8}{(s-s_0+4)^2 T_{s_0}}\left(\sum_{j=1}^{s_0} \frac{5C\sigma^2 2^{j-1}}{LB_j} + \sum_{j=s_0+1}^s \frac{5C\sigma^2 T_{s_0}}{4Lb'\alpha_j^2}\right) \\
&\leq \frac{16D_0}{(s-s_0+4)^2 m} + \frac{16}{(s-s_0+4)^2}\left(2^{-s_0}\sum_{j=1}^{s_0}(\frac{4}{3})^{j-1}\frac{5C\sigma^2}{Lb_1} + \sum_{j=s_0+1}^s \frac{5C\sigma^2(j-s_0+4)^2}{32Lb'}\right) \\
&\leq \frac{16D_0}{(s-s_0+4)^2 m} + \frac{16}{(s-s_0+4)^2}(\frac{2}{3})^{s_0}\frac{15C\sigma^2}{Lb_1} + \frac{5C\sigma^2(s-s_0)}{2Lb'} \\
&= \frac{\epsilon}{2} + \frac{\epsilon}{4} + \frac{\epsilon}{4} \quad s > s_0.
\end{aligned}$$

where the last equality holds when $s = s_0 + \sqrt{\frac{32D_0}{m\epsilon}} - 4$, $b_1 = (\frac{2}{3})^{s_0}\frac{30C\sigma^2 m}{LD_0}$ and $b' = \frac{10C\sigma^2(s-s_0)}{L\epsilon}$.

In this case, Varag needs to run at most run at most $S_h := s_0 + \sqrt{\frac{32D_0}{m\epsilon}} - 4$ epochs. Hence, the sampling complexity (number of calls to the SFO oracle) is bounded by

$$\begin{aligned}
\sum_{s=1}^{S_l}(mB_s + T_s b_s) &\leq 2m\sum_{s=1}^{s_0} b_1(\frac{3}{2})^{s-1} + 2mb'(S_h - s_0) \\
&\leq 4mb_1(\frac{3}{2})^{s_0} + \frac{20mC\sigma^2(S_h - s_0)^2}{L\epsilon} = \mathcal{O}\left\{\frac{C\sigma^2 D_0}{L\epsilon^2}\right\}, \tag{B.9}
\end{aligned}$$

and the communication complexity (CC), if in the distributed machine learning case, is bounded by

$$\sum_{s=1}^{S_h}(m + T_s) \leq 2m(s_0 + S_h - s_0) = \mathcal{O}\left\{m\log m + \sqrt{\frac{mD_0}{\epsilon}}\right\}, \quad m < \frac{D_0}{\epsilon}. \tag{B.10}$$

The result of Theorem 4 follows immediately by combining these two cases. $\qquad\square$

## C   More numerical experiments

**Strongly convex problems with small strongly convex modulus.** We consider ridge regression models with a small regularizer coefficient ($\lambda$) given in the following form,

$$\min_{x \in \mathbb{R}^n} \{\psi(x) := \frac{1}{m}\sum_{i=1}^{m} f_i(x) + h(x)\} \text{ where } f_i(x) := \frac{1}{2}(a_i^T x - b_i)^2, h(x) := \lambda\|x\|_2^2. \quad \text{(C.11)}$$

Since the above problem is strongly convex, we compare the performance of Varag with those of Prox-SVRG[3] and Katyusha[1]. As we can see from Figure 1, Varag and Katyusha converges much faster than Prox-SVRG in terms of training loss. Although Varag and Katyusha perform similar in terms of training loss per gradient calls, Varag may require less CPU time to perform one epoch than Katyusha. In fact, Varag only needs to solve one proximal mapping per inner iteration while Katyusha requires to solve two for strongly convex problems.

Diabetes ($m = 1151$), ridge $\lambda = 10^{-6}$          Breast-Cancer-Wisconsin ($m = 683$), ridge $\lambda = 10^{-6}$

Figure 1: In this experiments, the algorithmic parameters for Prox-SVRG and Katyusha are set according to [3] and [1], respectively, and those for Varag are set as in Theorem 2.