[Reviews · NeurIPS 2019]

Reviewer 1



This paper proposes a novel randomized incremental gradient algorithm Varag for minimizing convex problems with finite sum structure. The algorithm efficiently incorporate the variance reduction technique in order to achieve the optimal convergence rate for convex or strongly convex problems. The algorithm is at the same time simple and rich, which admits many interesting features. For instance, the algorithm admits a non-uniform sampling strategy to handle the case that the Lipschitiz constant are unbalanced; also the algorithm is able to handle non-Euclidean case via Bregman distance. It is a great contribution to combine all these components together in a relatively simple algorithm. Moreover, extensions to stochastic setting and error bound condition are also included. Experiments on logistic regression or Lasso are conducted showing that the algorithm is competitive compared to existing accelerated algorithms. Overall, I am very positive about the paper, even though I have some minor concerns. One of the key feature of the proposed algorithm is to have a phase transition on the parameters. In particular, the parameters are more "aggressive" in the early iterations. And this leads to a very surprising result that the algorithm is converging linearly in the early iterations even in the convex but non-strongly convex setting. More precisely, the linear convergence holds roughly before all the individual function are visited once. Moreover, the convergence parameter in this phase only depends on m, the number individual functions. Could authors provide more intuitions/explanations on why a linear convergence rate is achievable in the early stages? Furthermore, in the early stages, the number of inner loop iterations T_s are increasing exponentially, this is quite similar to the stochastic minimization setting that we increase the sample size geometrically. Could authors provide some motivation of this choice? In the experiment, is the non-uniform sampling strategy used? How does it compared to the case of uniform sampling? EDIT after author's feedback: I thank authors for the clarification, I believe the paper deserves to be published.

Reviewer 2



Originality. The problem of finite-sum minimization is very popular and attains a substantial attention recent years. Despite to the number of successive works in this field, results on both accelerated and variance-reduced methods are still under investigation and required additional research. The authors proposes new accelerated stochastic gradient method with variance reduction with its global complexity analysis. This method achieves the currently known convergence bounds and has a number of nice features (among others): * Tackling both convex and strongly convex problems in an unified way; * Using only one composite (proximal) step per iteration; * Working with general Bregman distances; * Relatively easy in implementation and analysis. To the best of my knowledge, up-to-know there are no accelerated variance-reduced methods having all these advantages simultaneously. Quality. The paper contains a number of theoretical contributions with proofs. Two algorithms (Varag and Stochastic Varag) are proposed in a clear and self-contained way. A number of numerical experiments on logistic regression with other state-of-the-art optimization methods are made. Clarity. The results are written in a well-mannered and clear way. A small concern from my side would be to make the introduction part more structured. Especially, more concrete examples of the cases, when we can not handle strong convexity of $f$ (but still know the constant) would be interesting. It also would be good to add more discussion and examples to the text, related to "Stochastic finite-sum optimization" (Section 2.2.). Significance. The results of this work seems important and significant to the area of stochastic variance-reduced accelerated methods.

Reviewer 3



Strengths: 1. This is a well written paper with strong theoretical results of wide appeal to the ML community in general and the NeurIPS community in particular. Accelerated methods are of particular importance for ill-conditioned problems. This makes Varag (like Katyusha) robust to ill-conditioned problems. 2. The Varag method takes a decisive (albeit by no means final) step in the direction of unifying certain algorithmic aspects of variance reduced methods for finite sum optimization. The results seamlesly cover both the strongly convex and convex regimes, the well and ill conditioned regimes, and the low and high accuracy regimes. Moreover, I like the additional support of a prox-function to better deal with the geometry of the constraint set, and support for linear convergence under and error bound condition, and support for infinite-sum (i.e., stochastic) optimization. 3. Varag achieves all results directly, unlike existing methods which in some cases rely on reformulation or perturbation. 4. To the best of my knowledge, Varag seems to be first accelerated variance reduced method which achieves a linear rate without strong convexity (and not for over-parameterized models): under an error bound condition. Such results have only recently been obtained for non-accelerated variance-reduced (Qian et al; SAGA with arbitrary sampling, ICML 2019); under a quadratic functional growth condition. Perhaps this can be mentioned. Issues and recommendations: 1. In Eq 2.6 one should have something else instead of x. Which optimal solution x^* should be used? This does not matter for the term, but matters for the second term. 2. Some explanations/statements are misleading. a. For instance, when explaining the results summarized in Table 2, the text in lines 74-78 mentioned linear convergence. However, in the regime m >= D_0/epsilon the rate O(m log 1/epsilon) is not linear in epsilon. The same issue is in lines 171-174 when describing the results of Theorem 1. b. The same issue again arises in lines 199-201 when describing the results of Theorem 2. Indeed, in the third case m can be replaced by D_0/epsilon, and this a sublinear rate and not a linear rate. 3. A further step towards unification would be achieved if support for minibatching was offered. Almost no one uses non-minibatch methods in practice currently. Surely, one can assume that each f_i depends in multiple data-points, forming an effective minibatch; and this would be covered by the provided results. But this only offers a limited flexibility for the formation of minibatches. 4. The abstract says: Varag is the “first of its kind” that benefits from the strong convexity of the data fidelity term. This is very vague as it is not at all clear what “first of its kind” means. Make this more clear. There exists many randomized variance-reduced methods that can utilize strong convexity present in the data fidelity term (e.g., the SAGA paper mentioned above, among many others). 5. A key issue with the Varag method is its reliance on an estimate of the strong convexity or error bound constant mu (\bar mu). Admittedly, all accelerated method suffer from this issue. One of the ways to handle this is via restarting, as done by, for instance, Fercoq and Qu (Adaptive restart of accelerated gradient methods under local quadratic growth condition, 2017; Restarting the accelerated coordinate descent method with a rough strong convexity estimate, 2018). This fact should be mentioned explicitly – I was surprised that this is left to the reader to discover; especially since strong claims are made about the unified nature of the method. The method is unified, but NOT adaptive to the strong convexity constants in the way other (non-accelerated) variance reduced methods are; e.g., SAGA. I would expect to see this difference in some experiments. That is, I suggest that experiments are performed with unknown mu, so that Varag has a hard time setting its parameters. Consider well and ill conditioned problems and compare Varag and SAGA or SVRG. 6. In practice, Loopless SVRG (Dmitry Kovalev et al, Don’t jump through hoops and remove those loops: SVRG and Katyusha are better without the outer loop, 2019) works better than SVRG. Also, it is better able to utilize information contained in the data (and, as SVRG, is adaptive to mu - even better so as it does not rely on mu to set the outer loop size). I wonder how this variant would compare in practice to Varag in the well conditioned / big data case (i.e., in the case when Varag is supposed switch to the non-accelerated regime). I would want to see a number of experiments on various artificial and real data, with the aim to judge whether Varag is able to do as well. I expect Varag to suffer as it needs to know the condition number to sets its parameters. 7. Can Varag be extended to a loopless variant, such as Loopless SVRG? This should lead to both a simpler analysis, and a better method in practice. 8. What happens with the results of Theorem 4 as one varies the outer and inner minibatch sizes b_s and B_s? A commentary is needed I think. 9. The experimental evaluation is not particularly strong. I made some recommendations above. Moreover, I do not see how were the various parameters set in the experiments. The text says: as in theory. But you do not know some of the constants, such as mu. How was mu computed? Was the calculation included in the runtime when comparing against SVRG? Small issues and typos: The paper contains a relatively large (but not excessively large) number small issues and typos, which need to be fixed. Some examples: 1. 3: conditional number -> condition number 2. 12: function -> functions 3. 13: smooth convex functions with -> smooth and convex 4. 17: necessarily -> necessary 5. 20: becomes -> became 6. 23: connect -> connected 7. 38: number of components -> number of components is 8. 88: rates -> rate 9. 97: class -> classes 10. 106: in -> to 11. 116: You may wish to use the words ceiling and floor instead. 12. 126: to simple -> to be simple 13. 142: log(1/eps -> log(1/eps) 14. 150: computational -> computationally 15. 154: updates -> update 16. 155: non-euclidean -> non-Euclidean 17. 162: its -> the 18. 199: smooth finite-sum -> smooth convex finite-sum 19. 199: strongly convex modulus -> strong convexity modulus 20. 222: an unified result as we shown in Theorem 2 -> a unified result, as shown in Theorem 2, 21. 223: conditional number -> condition number ----------------- I've read the other reviews and the rebuttal. I am keeping my score - this is a good paper.

[Author Response · NeurIPS 2019]

We thank the reviewers for appreciating our work, and for their constructive suggestions to improve its quality.

*Response to Review #1:* **Intuition on linear rate:** Varag achieves linear convergence rate when $m \geq D_0/\epsilon$, and
sublinear rate when $m < D_0/\epsilon$, which relies on our selection of the inner loop size $T_s$. In our convergence analysis, we
notice that the convergence rate is roughly in the order of $1/T_s$ (see Lemma 7), hence, if $T_s$ increases exponentially,
we can achieve linear convergence rate. Intuitively, it is reasonable to always increase $T_s$ in order to avoid the full
gradient computation when $m$ is very large, i.e., $m \geq D_0/\epsilon$. It then stops increasing $T_s$ when $T_s = m$, since the cost
of full gradient computation is comparable to that required by $m$ inner loops. We will add such discussions in the
text. **Sampling method in experiments:** We use uniform sampling strategy to select $f_i$ in all experiments. Indeed,
theoretically the sampling distribution can be non-uniform, i.e., $q_i = L_i / \sum_{i=1}^{m} L_i$, which results in the optimal constant
$L = \frac{1}{m} \sum_{i=1}^{m} L_i$ appearing in the convergence results. A uniform sampling, e.g., $q_i = \max L_i$, will lead to a constant
factor slightly larger than $L$. Note that $L_i$ can be estimated by performing maximum singular value decomposition of
the Hessian. This is computationally efficient because only a rough estimation suffices. We appreciate the reviewers'
comments and will add corresponding experiments and discussions in the experiment section.

*Response to Review #2*: **Introduction section:** We will add more examples and discussions on cases of strong convexity
of $f$ and stochastic finite-sum problems. **$D_0$ in Table 2:** We will fix the footnote for $D_0$. **Sampling distribution:** See
response to review #1 about a similar question. **Relation with other accelerated methods:** We pointed out in the
footnote 1 that Catalyst requires restarting to achieve the optimal convergence rates, Katyusha needs to add perturbations
to achieve optimal rates for smooth problems. We will expand these discussions and put them into main text. Note that
we also compare Varag with Katyusha in details after we present Varag in Alg. 1.

*Response to Review #3*: 1. Thanks for pointing out this typo. $x$ should be $x^*$ in Equation (2.6).
2. We say that $O(m \log 1/\epsilon)$ is linear but not sublinear w.r.t. $\epsilon$. We cannot replace $m$ by $D_0/\epsilon$ because it leads to a
too optimistic bound. Moreover, $m$ is a constant independent of $\epsilon$, and roughly every $m$ gradient computations will
increase 1 digit of accuracy, so we call it a linear rate. Indeed we admit that an $O((D_0/\epsilon) \log 1/\epsilon)$ bound would be
better than $O(m \log 1/\epsilon)$ if $m > D_0/\epsilon$. We will add such discussions in respective places in the main text.

3. Thanks for this suggestion. Indeed one can assume each individual $f_i$ is associated with a minibatch instead of a
single piece of data. For the more general minibatch version, one can replace $G_t = (\nabla f_{i_t}(\underline{x}_t) - \nabla f_{i_t}(\tilde{x}))/(q_{i_t} m) + \tilde{g}$
(Line 7 of Algorithm 1) by $G_t = \frac{1}{b} \sum_{i_t \in S_b} (\nabla f_{i_t}(\underline{x}_t) - \nabla f_{i_t}(\tilde{x}))/(q_{i_t} m) + \tilde{g}$ with $|S_b| = b$ and adjust the appropriate
parameters to obtain the minibatch Varag. We expect that the minibatch Varag will obtain the parallel linear speedup of
factor $b$ if minibatch size $b \leq \sqrt{m}$. We will incorporate such analysis into the revision of the paper.

4. We will update it as "Varag is the first accelerated randomized incremental gradient method that benefits from the
strong convexity of the data-fidelity term to achieve the optimal linear convergence" to be more accurate.

5. Our Varag method is not adaptive and we will mention this explicitly in the later version. Note that the adaptivity of
hyperparameters, i.e., smooth parameter $L$ and strongly convex parameter $\mu$ (Varag only needs these two hyperparame-
ters), is not the focus of our current Varag method. Varag uses a unified step-size policy to unify the convex problems
with or without strong convexity, and directly achieve the best convergence rate for non-strongly convex problems. The
adaptivity of hyperparameters is a good property for an algorithm, and we leave this as an interesting future extension
of our work. We appreciate the reviewer's question and will add clarification into the revision.

6. Thanks for pointing out the recently Loopless SVRG paper [KHR2019] which removes the outer loop of SVRG
by computing the full gradient with a small probability in each iteration. In the well conditioned/ big data case, our
Varag switches to non-accelerated regime and achieves a linear convergence rate. We would like to point out that in
non-accelerated regime, Varag, similar to Loopless SVRG, only needs to know $L$ and does not require the knowledge
of $\mu$ to set its parameters. Thus we believe that Varag will still work well in this case. We will some discussions about
Varag's properties in this regime, as well as Loopless SVRG.

7. After briefly reading Loopless SVRG [KHR2019], we feel that Varag can also be possibly generalized into a loopless
version similar to Loopless SVRG. We will discuss possible extensions of this method in the revised version.

8. We provide the theoretic suggestion of $b_s$ and $B_s$ by minimizing the stochastic gradient complexity: $\sum_s mB_s +$
$\sum_s T_s b_s$. One can use other values for $b_s$ and $B_s$ and Varag can still converge to a stochastic $\epsilon$-solution, but it may lead
to a worse stochastic gradient complexity than our theoretic guarantee. We will add such discussions into the text.

9. Thanks for the constructive suggestions. We will try to add more experiments. Regarding the parameters, we only
need two hyperparameters (i.e., the smooth parameter $L$ and strongly convex parameter $\mu$) to set all parameters in our
experiment. We first use singular value decomposition (SVD) for the Hessian to compute $L$ and $\mu$ at the beginning for
all algorithms (this step is not included in the performance comparison). Then we use them to run all algorithms and
compare their performance w.r.t. gradient computation. We will specify more details of the experiments and parameters
setting in the revised version.

10. All typos will be addressed in the revised paper.

[Meta-Review · NeurIPS 2019]

There is a clear consensus among the reviewers that the contribution is interesting and that the paper is suitable to publication. The area chair agrees with their assessment and follows their recommendation.